# Ultrasmall metal alloy nanozymes mimicking neutrophil enzymatic cascades for tumor catalytic therapy

Xiangqin Meng[1,8], Huizhen Fan [1,8], Lei Chen[1,2,8], Jiuyang He[3,8], Chaoyi Hong[1,4], Jiaying Xie[1,4], Yinyin Hou[1,4], Kaidi Wang[1,4], Xingfa Gao [5], Lizeng Gao [1,4,6], Xiyun Yan [1,4,6,7,9] ✉ & Kelong Fan [1,4,6,7,9] ✉

Developing strategies that emulate the killing mechanism of neutrophils, which involves the enzymatic cascade of superoxide dismutase (SOD) and myeloperoxidase (MPO), shows potential as a viable approach for cancer therapy. Nonetheless, utilizing natural enzymes as therapeutics is hindered by various challenges. While nanozymes have emerged for cancer treatment, developing SOD-MPO cascade in one nanozyme remains a challenge. Here, we develop nanozymes possessing both SOD- and MPO-like activities through alloying Au and Pd, which exhibits the highest cascade activity when the ratio of Au and Pd is 1:3, attributing to the high d-band center and adsorption energy for superoxide anions, as determined through theoretical calculations. The $Au_1Pd_3$ alloy nanozymes exhibit excellent tumor therapeutic performance and safety in female tumor-bearing mice, with safety attributed to their tumor-specific killing ability and renal clearance ability caused by ultrasmall size. Together, this work develops ultrasmall AuPd alloy nanozymes that mimic neutrophil enzymatic cascades for catalytic treatment of tumors.

Evolution has resulted in the development of a robust immune system, which can serve as a source of inspiration and template for the exploration of emerging and innovative strategies for tumor therapy. Neutrophils are the first line of defense of the innate immune system[1], and play an important role in resisting microbial infections[2] and combatting tumors[3]. Neutrophils engulf microorganisms or tumor cell membranes to generate intracellular phagosomes that subsequently fuse with the azurophilic granules in the cytoplasm, causing the release of myeloperoxidase (MPO) from the granules, thereby triggering the activation of multiple natural enzymes mediated cascade reactions for effective killing function. Essentially, a cascade reaction occurs where

the superoxide anion ($O_2^{\cdot-}$) in the phagosome undergoes disproportionation into hydrogen peroxide ($H_2O_2$) and oxygen ($O_2$) through the catalysis of superoxide dismutase (SOD). The resulting $H_2O_2$ is utilized as a substrate by MPO along with chloride ion ($Cl^-$), resulting in the generation of potent oxidizing agents hypochlorous acid (HClO) and singlet oxygen ($^1O_2$). These reactive oxygen species (ROS) have the capability to destroy proteins, nucleic acids, and other vital biomolecules, leading to destructive effects on microorganisms or tumor cells. Compared to another commonly used ROS, hydroxyl radicals ($\cdot OH$), $^1O_2$ exhibits superior half-life and diffusion distance[4]. The half-life of $^1O_2$ (1-4 μs) is 1000-4000 times longer than that of $\cdot OH$

[1]CAS Engineering Laboratory for Nanozyme, Key Laboratory of Biomacromolecules (CAS), CAS Center for Excellence in Biomacromolecules, Institute of Biophysics, Chinese Academy of Sciences, Beijing 100101, PR China. [2]Institute of Translational Medicine, Medical College, Yangzhou University, Yangzhou 225001, PR China. [3]Experimental Center of Advanced Materials, School of Materials Science & Engineering, Beijing Institute of Technology, Beijing 100081, PR China. [4]University of Chinese Academy of Sciences, Beijing 101408, PR China. [5]National Center for Nanoscience and Technology, Beijing 100190, PR China. [6]Nanozyme Medical Center, School of Basic Medical Sciences, Zhengzhou University, Zhengzhou 450052, PR China. [7]Nanozyme Laboratory in Zhongyuan, Zhengzhou 451163 Henan, PR China. [8]These authors contributed equally: Xiangqin Meng, Huizhen Fan, Lei Chen, Jiuyang He. [9]These authors jointly supervised this work: Xiyun Yan, Kelong Fan. ✉e-mail: yanxy@ibp.ac.cn; fankelong@ibp.ac.cn

(1 ns), and its diffusion distance (30 nm) is 30 times greater than that of ·OH (1 nm).

Using extracted natural enzymes to simulate neutrophil enzymatic cascade reactions to effectively generate HClO and $^1O_2$ has become a promising strategy for cancer therapy[5,6]. However, due to the fact that the chemical nature of natural enzymes is protein, they typically exhibit poor stability and are prone to structural changes and inactivation in acidic, alkaline, and thermal environments. In addition, the extraction process of natural enzymes is complex and costly. Furthermore, there is an issue of immunogenicity in the application of natural enzymes in the body. These factors limit the clinical development and application of natural enzymes as drugs[7,8].

Nanozymes provide materials for the development of enzyme-catalyzed therapies[9,10]. The nanostructure of nanozymes endows them not only with efficient catalytic function, but also with greater stability and easier scalability than natural enzymes. This enables nanozymes to replace and even surpass natural enzymes, making them widely applicable in multiple fields[11–13]. Nanozymes have demonstrated great potential in regulating ROS for tumor catalytic therapy and antimicrobial applications. However, it is worth noting that the therapeutic efficacy of these nanozymes can be limited due to their low affinity for $H_2O_2$ and the typically low levels of $H_2O_2$ in the tumor microenvironment and inside microbes[14]. In a previous report[15], a single-atom nanozyme was developed, capable of catalyzing the production of $O_2^{·-}$ and HClO from $H_2O_2$ and $Cl^-$. This effectively promoted the elimination of drug-resistant bacteria and the healing of bacterial infections. However, lacking SOD-like activity, the nanozyme was unable to catalyze the generation of $H_2O_2$ as a substrate for MPO, thus requiring additional $H_2O_2$ during the treatment. When treating tumors, the restricted level of $H_2O_2$ in the tumor microenvironment (less than 0.1 mM)[6] makes it also necessary for additional $H_2O_2$, not only increasing the complexity of treatment, but also raising the difficulty of subsequent clinical translation. Therefore, developing nanozymes that possess dual enzyme-like activities of SOD and MPO to simulate the neutrophil enzymatic cascades for tumor therapy is highly valuable albeit quite challenging.

In recent years, noble metal nanozymes have attracted considerable attention due to their remarkable catalytic activity[16]. Alloying further offers an opportunity to greatly extend the range of properties exhibited by noble metal nanozymes[17]. In particular, we are intrigued by the AuPd alloys, which show electronic structures that are distinct from those of the pure metals due to the differing atomic electron configurations and electronegativities of Au and Pd[18].

In this work, we adjust the alloy ratio to develop a series of ultrasmall metal nanozymes: Au, $Au_3Pd_1$, $Au_2Pd_2$, $Au_1Pd_3$, Pd, and find that these nanozymes exhibit alloy ratio-dependent SOD- and MPO-like activities, among which the $Au_1Pd_3$ alloy nanozymes present the highest cascade activity. Moreover, we use theoretical calculations to elucidate the catalytic mechanisms of these nanozymes and find that the high d-band center and adsorption energy for $O_2^{·-}$ contribute to the SOD-MPO cascade activity. Importantly, the $Au_1Pd_3$ alloy nanozymes successfully mimic the SOD-MPO cascade enzymatic killing function of neutrophils and cause DNA damage and cell apoptosis by generating HClO amd $^1O_2$, thereby demonstrating excellent tumor therapeutic effect (Fig. 1). In addition, the $Au_1Pd_3$ alloy nanozymes also exhibit good in vivo safety owing to their tumor-specific cytotoxicity and ultrasmall size of less than 6 nm that facilitates their clearance through the kidneys and prevents long-term accumulation in the body. Together, we successfully develop ultrasmall AuPd alloy nanozymes with SOD- and MPO-like activities to mimic neutrophil enzymatic cascades for efficient and safe tumor catalytic therapy.

## Results
### Synthesis and characterization of ultrasmall AuPd nanozymes
Ultrasmall AuPd alloy nanozymes were prepared through a modified method based on $P(CH_2OH)_4Cl$ (tetrakis (hydroxymethyl)

phosphonium chloride, THPC)[19,20] (Fig. 1). The alloy ratio was regulated by adjusting the proportions of metal precursors. The surface of the ultrasmall AuPd alloy nanozymes was coated with SH-PEG-OMe through strong coordination between metal and thiol (SH) to maintain stability and dispersibility, as well as reduce non-specific adsorption of proteins[21,22]. High-resolution transmission electron microscopy (HRTEM) images demonstrated the spherical morphology and uniform distribution of the synthesized five nanozymes, all exhibiting a diameter range of 2-3 nm. (Fig. 2a). The Pd content in the five nanozymes was determined by an inductively coupled plasma optical emission spectrometer (ICP-OES). The results showed that the five nanozymes contained increasing amounts of Pd, indicating that we successfully obtain five nanozymes with different alloy ratios (Fig. 2b). These nanozymes were labeled as Au, $Au_3Pd_1$, $Au_2Pd_2$, $Au_1Pd_3$, and Pd, corresponding to their respective alloy compositions. The coverage of SH-PEG-OMe was confirmed by Fourier transform infrared (FTIR), which showed strong bands at 2890 $cm^{-1}$ and 1112 $cm^{-1}$ related to the C-H and C-O stretching vibrations of PEG main chain, respectively (Fig. 2c)[23]. The interaction between SH-PEG-OMe and AuPd alloy nanozymes through the Au-S bond or Pd-S bond was verified by the appearance of peaks at about 300 $cm^{-1}$ in the Raman spectra (Fig. 2d)[24,25]. The structures of ultrasmall AuPd alloy nanozymes were determined by powder X-ray diffraction (XRD) (Fig. 2e). The XRD patterns of Au and Pd nanozymes can be indexed to the face-centered cubic Au and Pd, and the peaks of $Au_3Pd_1$, $Au_2Pd_2$ and $Au_1Pd_3$ alloy nanozymes showed a shift from Au (PDF#04-0784) to Pd (PDF#46-1043), demonstrating the formation of alloy structures. X-ray photoelectron spectroscopy (XPS) was used to analyze the effect of alloying on the chemical states of AuPd alloy nanozymes. It can be seen from the XPS spectra that all five nanozymes contained related metal elements and C, O elements that constitute SH-PEG-OMe. Additionally, Na was present in the nanozymes due to the addition of NaOH during the synthesis process (Supplementary Fig. 1). The valence states of Au in Au nanozymes encompassed $Au^0$, $Au^{1+}$, and $Au^{3+}$, as illustrated in Fig. 2f. Similarly, Pd nanozymes exhibited the valence states of $Pd^0$ and $Pd^{2+}$. These results suggest that the surface of Au nanozymes undergo the adsorption of $Au[Cl_x(OH)_{4-x}]^-$ species, while Pd nanozymes experience the adsorption of $Pd[Cl_x(OH)_y]^{2-x-y}$ species[26,27]. Pd in AuPd alloy nanozymes would exhibit a certain degree of electropositivity because of the higher electronegativity of Au[28,29]. As the proportion of Pd on the surface increased from Au to $Au_3Pd_1$ nanozymes, the percentage of $Au^{3+}$ increased because the positively charged Pd adsorbed more $Au[Cl_x(OH)_{4-x}]^-$, in which Au refers to $Au^{3+}$. The coverage of SH-PEG-OMe made the five nanozymes stable in solution, with average hydrodynamic diameters of 3.8−5.3 nm (Fig. 2g). Their ultrasmall size (less than 6 nm) gives them the potential for renal clearance[30]. Above results show that we successfully synthesized nanozymes with ultrasmall size and different alloy ratios.

### Imitation of the cascade catalysis of neutrophils by AuPd nanozymes
It has been reported that alloying can finely tune the catalytic activity of metals[18]. Thus, we investigated the enzyme-like activities of five AuPd nanozymes and explored the relationship between the activity and alloy ratio. Interestingly, we found that these nanozymes exhibited both SOD- and MPO-like activities simultaneously, and their activities were closely related to the alloy ratio. The SOD-like activity was determined by a SOD assay kit, and normalized by the molar concentration to compare the difference between nanozymes with different degrees of alloying. As shown in Fig. 3a, the SOD-like activity of either Au or Pd nanozymes was low, and alloying could increase their activity. $Au_1Pd_3$ alloy nanozymes exhibited the highest SOD-like activity, with a specific activity of about 7300 U $mg^{-1}$, which is much higher than that of natural SOD[31]. It is supposed that as the Pd content in the alloy increased, the SOD-like activity of AuPd

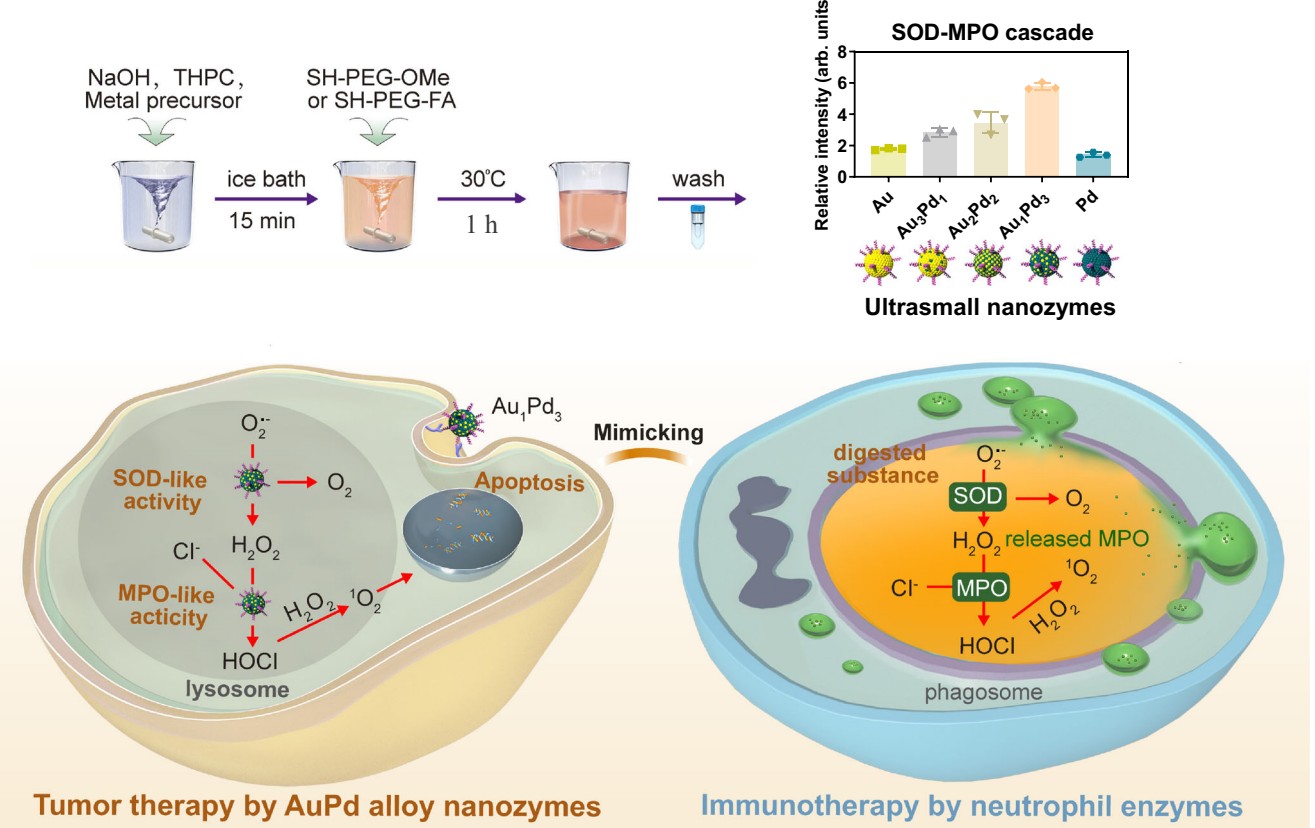

**Fig. 1 | Schematic illustration of ultrasmall AuPd alloy nanozymes mimicking the SOD-MPO enzymatic cascade killing function of neutrophils for tumor catalytic therapy.** Ultrasmall metal nanozymes with different alloy ratios are synthesized. These nanozymes exhibit alloy ratio-dependent SOD and MPO-like activities, among which the $Au_1Pd_3$ alloy nanozymes present the highest cascade activity. The $Au_1Pd_3$ alloy nanozymes mimic the SOD-MPO cascade enzymatic therapy performed by neutrophils, demonstrating effective tumor therapy by inducing DNA damage and cell apoptosis via producing HClO and $^1O_2$.

nanozymes should exhibit a pattern of initially increasing and then decreasing, with the corresponding peak ratio being the optimal ratio. In order to determine the alloy ratio of nanozymes with the optimal SOD-like activity, we further prepared the $Au_{1.3}Pd_{8.7}$ alloy nanozymes, with the Pd content in the alloy determined to be 87% by the ICP-OES analysis. As shown in Supplementary Fig. S2, the $Au_{1.3}Pd_{8.7}$ nanozymes consisted of uniformly distributed small particles with a hydrated particle size of 3.8 nm, which was consistent with the other five nanozymes. The results of the SOD activity assay demonstrated that the $Au_{1.3}Pd_{8.7}$ alloy nanozymes possessed lower activity than the $Au_1Pd_3$ alloy nanozymes, but higher activity than the Pd nanozymes. These results suggest that the alloy nanozymes with an Au:Pd ratio of 1:3 exhibit the optimal SOD-like activity. The MPO-like activities of AuPd nanozymes were detected by monitoring the conversion of monochlorodimedon (MCD) to dichlorodimedon[32], using the change in absorbance of MCD at 290 nm as an indicator (Supplementary Fig. 3). The statistical results showed that the MPO-like activities of AuPd nanozymes increased with the increase of the proportion of Pd (Fig. 3b). The high SOD- and MPO-like activities of the AuPd alloy nanozymes also benefit from their ultrasmall size, as we found that the ultrasmall-sized AuPd alloy nanozymes exhibited higher enzymatic activities compared to the large-sized AuPd alloy nanozymes (Supplementary Fig. 4), which may be related to their higher specific surface area, potentially exposing more active sites. Moreover, the SOD- and MPO-like activities of the AuPd alloy nanozymes remained stable over a wide range of temperatures from 25 to 80 °C (Supplementary Fig. 5), which is due to the nanostructural stability of AuPd alloys as inorganic nanomaterials. To investigate the potential cascade reaction involving the SOD- and MPO-like activities

of AuPd alloy nanozymes, the experimental setup deliberately excluded the inclusion of the MPO substrate, $H_2O_2$, within the reaction system. Instead, pyrogallol was employed as an $O_2^-$ generator[33] to facilitate SOD-like activity, thereby generating $H_2O_2$ as a product. This approach aimed to assess whether a cascade reaction could be initiated and sustained solely through the interplay of SOD- and MPO-like activities within the AuPd alloy nanozymes. Subsequently, we conducted an analysis to detect the formation of HClO, a product associated with MPO activity, as well as the additional product $^1O_2$ to determine the occurrence of the cascade reaction (Fig. 3c). Due to the fact that pyrogallol could affect the absorption of MCD, we detected HClO by subtracting the fluorescence of the 2-[6-(4'-hydroxy)phenoxy-3H-xanthen-3-on-9-yl]benzoic acid (HPF) probe from the fluorescence of the 2-[6-(4'-amino)phenoxy-3H-xanthen-3-on-9-yl]benzoic acid (APF) probe[34]. The APF probe reacts with HClO and ˙OH, while the HPF probe only reacts with ˙OH. In addition, 9,10-anthracenediyl-bis (methylene) dimalonic acid (ABDA) as the $^1O_2$-specific probe was used to detect the $^1O_2$ generation through the cascade reaction[35]. The results showed that the generated product $H_2O_2$ of SOD was used as the substrate of MPO, indicating that the SOD- and MPO-like activities of AuPd alloy nanozymes successfully form a cascade reaction. Among five nanozymes, $Au_1Pd_3$ alloy nanozymes exhibited the highest SOD-MPO-like cascade activity and produced the highest amounts of HClO (Fig. 3d and Supplementary Fig. 6) and $^1O_2$ (Fig. 3e), which may be related to its highest SOD-like activity. The above results indicate that AuPd alloy nanozymes, especially $Au_1Pd_3$, can simulate the neutrophil SOD-MPO enzyme-catalyzed cascade reaction, transforming $O_2^-$ and $Cl^-$ into HClO and $^1O_2$.

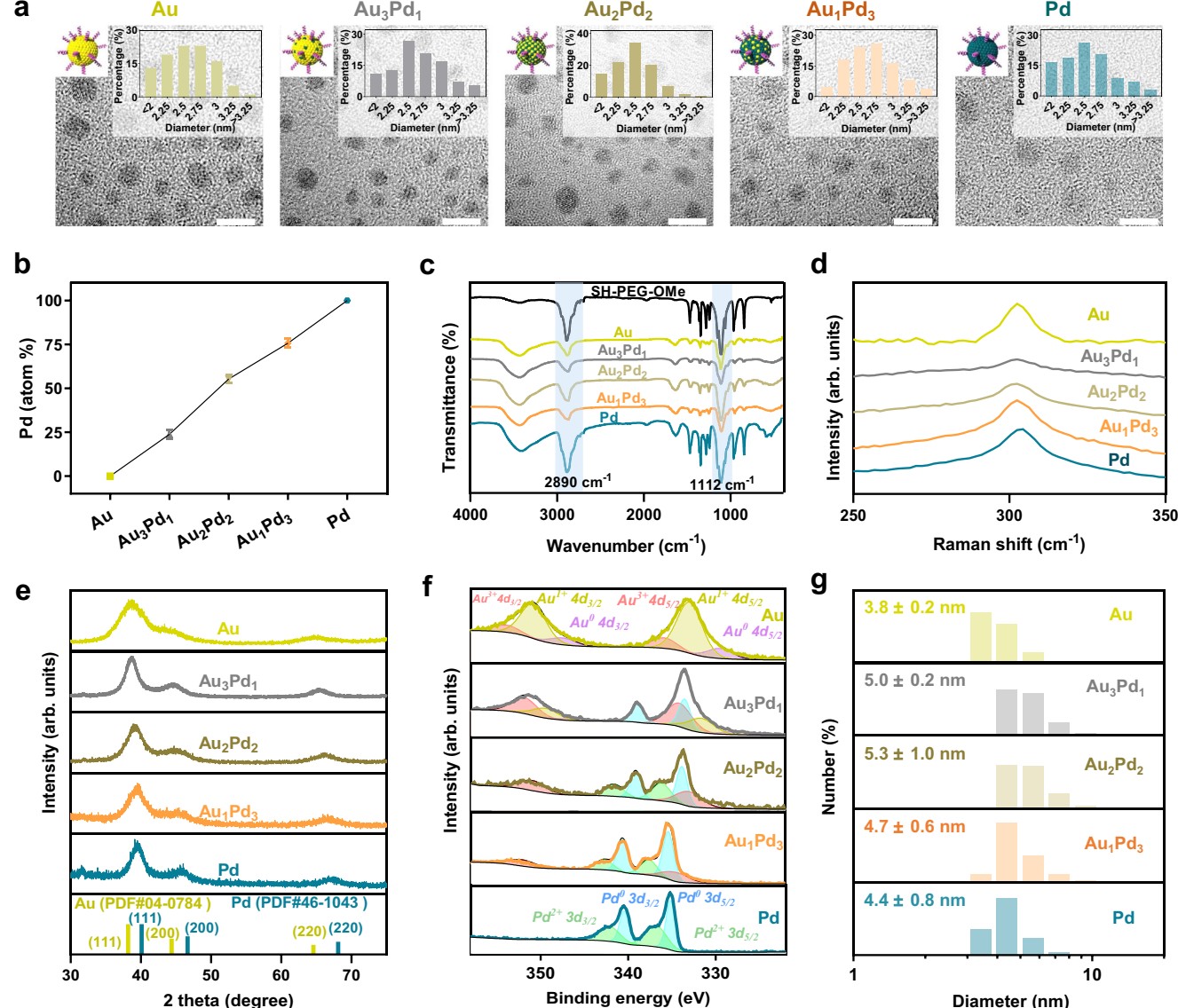

**Fig. 2 | Synthesis and characterization of ultrasmall AuPd nanozymes. a** HRTEM images, size distribution (n = 100 nanoparticles) and schematic illustration of five ultrasmall nanozymes. Scale bars = 5 nm. **b** The Pd content in five AuPd alloy nanozymes determined by ICP-OES (n = 3 independent experiments). **c** FTIR spectra of five AuPd nanozymes and SH-PEG-OMe. **d** Raman spectra of five AuPd nanozymes. **e** XRD patterns and (**f**) XPS spectra of five AuPd nanozymes. **g** Hydrodynamic diameter distribution of five ultrasmall nanozymes. All data are presented as mean ± STD. Source data are provided as a Source Data file.

## Theoretical calculations of the SOD-MPO-like cascade catalysis of AuPd nanozymes

In AuPd catalysts, the ligand, strain, and ensemble effects play vital roles in determining catalytic activity. The ensemble effect refers to modifications in the surface properties that occur due to a direct alteration in the atomic ensemble constituents. The ligand and strain effects describe the tuning of the surface electronic structure in a specific surface ensemble and the changes in bond lengths of catalysts due to differences in the lattice constants of the components, respectively. Previous researches have indicated that the ensemble effect plays a significant role in the selectivity of catalytic reactions[36,37]. Furthermore, the ensemble effect exerts a greater influence on the adsorbate binding on the AuPd surface, surpassing the effects of ligand and strain[38,39]. Our experimental results demonstrated that the ensemble effect that was related to the ratio of the two metals in AuPd alloy nanozymes significantly affected their SOD-MPO-like cascade catalysis. Subsequently, we performed density functional theory (DFT) calculations to investigate the catalytic mechanism.

The d-band center theory states that, the d-band center for various metals is an indicator to explain the adsorption energy trends for a given adsorbate: the higher the d-states are in energy relative to the Fermi level, the more empty the anti-bonding states and the larger the adsorption energy, which has been widely used to explain the relative reactivity of metal surfaces[40–42]. The (111) facets of Au, Pd, and their alloys were selected accounting for catalytic activities in our calculations due to their more energetically stable configurations than the other facets[43,44]. Moreover, the (111) facets were also the predominant facets in the synthesized nanozymes, as shown by XRD patterns in Fig. 2e. To rationalize the catalytic activity order, first, the d-band centers for the metal (111) facets of Au, Pd and their alloys were calculated as presented in Fig. 4a. The d-band center order was Pd (111) > Au₁Pd₃ (111) > Au₂Pd₂ (111) > Au₃Pd₁ (111) > Au (111), which was basically consistent with the experimental SOD-MPO-like cascade catalytic activity (Fig. 2d, Au₁Pd₃ > Au₂Pd₂ > Au₃Pd₁ > Au > Pd) except the order of Pd (111) and Au₁Pd₃ (111), ascribing to the approximation of the molecular energy levels of $O_2^-$ and the ignorance of the

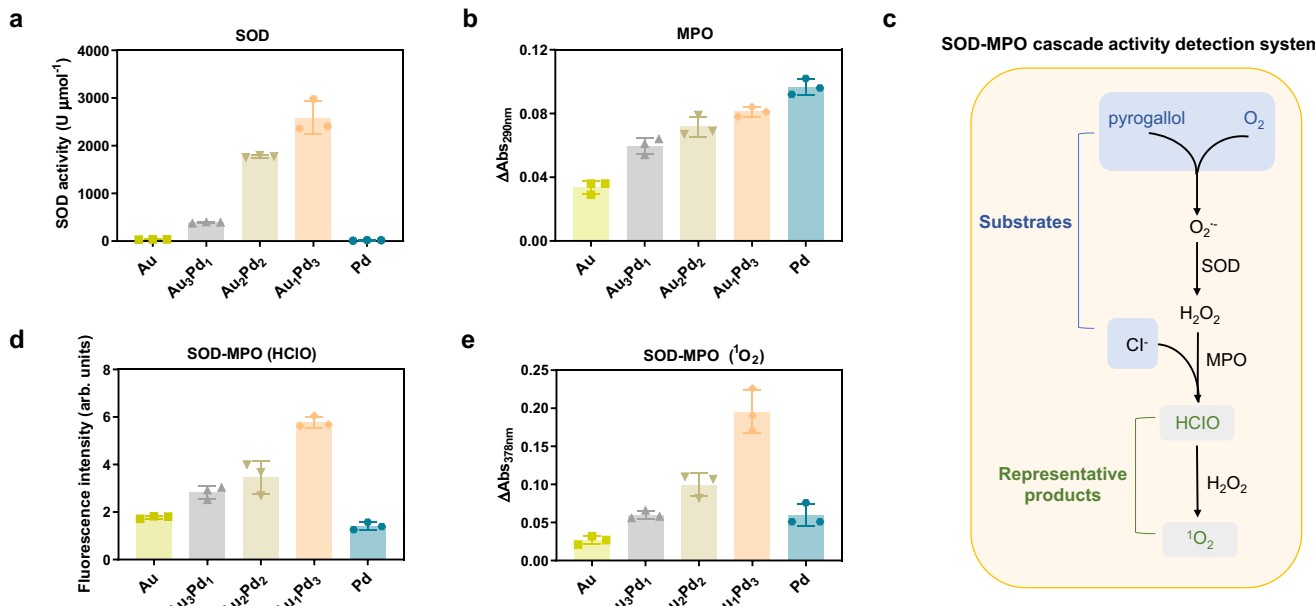

**Fig. 3 | SOD-like and MPO-like activities of five AuPd nanozymes. a** SOD- and **b** MPO-like activity of five AuPd nanozymes ($n = 3$ independent experiments). **c** Method for detecting the SOD-MPO-like cascade activity. **d** Production of HClO and **e** $^1O_2$ by the SOD-MPO-like cascade activity ($n = 3$ independent experiments). All data are presented as mean ± STD. Source data are provided as a Source Data file.

characteristics of molecular orbitals and their occupied states of $O_2^{\cdot-}$ in d-band center theory. To verify the difference of the cascade catalysis between Pd (111) and $Au_1Pd_3$ (111), we calculated the reactants, products, intermediates and transition states in the possible reaction pathway of the SOD-MPO-like cascade catalytic activity for $O_2^{\cdot-}$ and $Cl^-$ on Pd (111) and $Au_1Pd_3$ (111) surfaces. The $O_2^{\cdot-}$ radical is a Brønsted base with $pK_b = 9.12$[45], so it can easily capture a proton forming $HO_2^{\cdot}$ radical in acidic solution as illustrated by the following equation:

$$O_2^{\cdot-} + H^+ \rightarrow HO_2^{\cdot} \tag{1}$$

The HCl molecule was selected as one of the substrates of MPO-like activity in acidic solution. We then studied the cascade catalysis of the decomposition reactions of two $HO_2^{\cdot}$ radical and one HCl molecule on Pd (111) and $Au_1Pd_3$ (111) surfaces (Eq. (2) as the whole SOD-MPO-like cascade catalytic reaction), and the following Eqs. ((3)–(5)) serve as a plausible mechanism:

$$2HO_2^{\cdot} + HCl \rightarrow O_2 + H_2O + HClO \tag{2}$$

$$HO_2^{\cdot} + metal\,(111)\,facet \rightarrow HO_2^{\cdot\,*} \tag{3}$$

$$HO_2^{\cdot} + HO_2^{\cdot\,*} \rightarrow O_2 + H_2O_2 \rightarrow O_2 + H_2O + O^* \tag{4}$$

$$HCl + O^* \rightarrow HClO^* \tag{5}$$

Asterisk (*) is used to mark the species adsorbed on $Au_1Pd_3$ (111) and Pd (111) facets. On the basis of the energetically favorable adsorption configurations as listed in Fig. 4b, the top adsorption configuration of $HO_2^{\cdot}$ radical on metal (111) surface was most stable and selected as the initial structure to construct the proposed reaction pathways of the SOD-MPO-like cascade catalytic activity on $Au_1Pd_3$ (111) and Pd (111) surfaces. The corresponding dissociation and transition state configurations were located in Fig. 4c and e, respectively, and the corresponding potential energy profiles were present in Fig. 4d, f, respectively.

As shown in Fig. 4c and e, there were six steps from the adsorption of the first $HO_2^{\cdot}$ radical on metal (111) surface to the formation of HClO, including five stable states and one transition state in the reaction pathway of SOD-MPO-like cascade activity. The whole cascade catalytic cycle (Eq. (2)) consisted of three dominant elementary reactions (Eqs. (3)–(5)): (1) adsorption of $HO_2^{\cdot}$ radical on metal (111) surface ($HO_2^{\cdot} + metal\,(111)\,surface \rightarrow HO_2^{\cdot*}$); (2) SOD-MPO-like cascade catalytic activity forming $O^*$ ($HO_2^{\cdot} + HO_2^{\cdot}\,^* \rightarrow O_2 + H_2O_2 \rightarrow O_2 + H_2O + O^*$); (3) MPO-like activity forming HClO ($HCl + O^* \rightarrow HClO^*$).

To further determine the final product of two $HO_2\cdot$ radicals on $Au_1Pd_3$ (111) and Pd (111) surfaces, we evaluated the thermal stability of two $HO_2\cdot$ radicals at 300 K. The abinitio molecular dynamics (AIMD) simulations were carried out using a canonical ensemble with a Nosé−Hoover heat bath scheme[46]. The simulation time was 20 ps. The AIMD simulations (Fig. 4g, h) indicate that the $HO_2\cdot\,^*$ can thermodynamically convert to $O^*$ after addition of the second $HO_2\cdot$ radical. These results confirm the rationality of the process from state 2 to state 3 in Fig. 4d, f. The potential energy profiles show that there was almost no difference between $Au_1Pd_3$ (111) and Pd (111) surfaces in the process from state 3 to state 5, which was consistent with the minor difference in MPO-like activity between $Au_1Pd_3$ and Pd nanozymes in the experimental results (Fig. 2b). From Fig. 2, it can be seen that the SOD-like activity trend for five nanozymes was consistent with the trend of SOD-MPO-like cascade activity, indicating that SOD-like activity is the key to the overall cascade activity. Compared with Pd (111) surface, the binding energy between $HO_2\cdot$ and $Au_1Pd_3$ (111) surface was much lower (−1.61 eV vs. −1.30 eV), which indicates that the $Au_1Pd_3$ (111) surface have stronger ability to capture $HO_2\cdot$ free radicals. In the Michaelis−Menten equation, $K_M$ is the dissociation constant of the reaction in Eq. (3):

$$HO_2^{\cdot} + metal\,(111)\,facet \rightarrow HO_2^{\cdot\,*}, \Delta G_m^o \tag{6}$$

where the $\Delta G_m^o$ is the change of standard Gibbs free energy. According to the van't Hoff equation,

$$K_M^o = e^{\Delta G_m^o/RT} \tag{7}$$

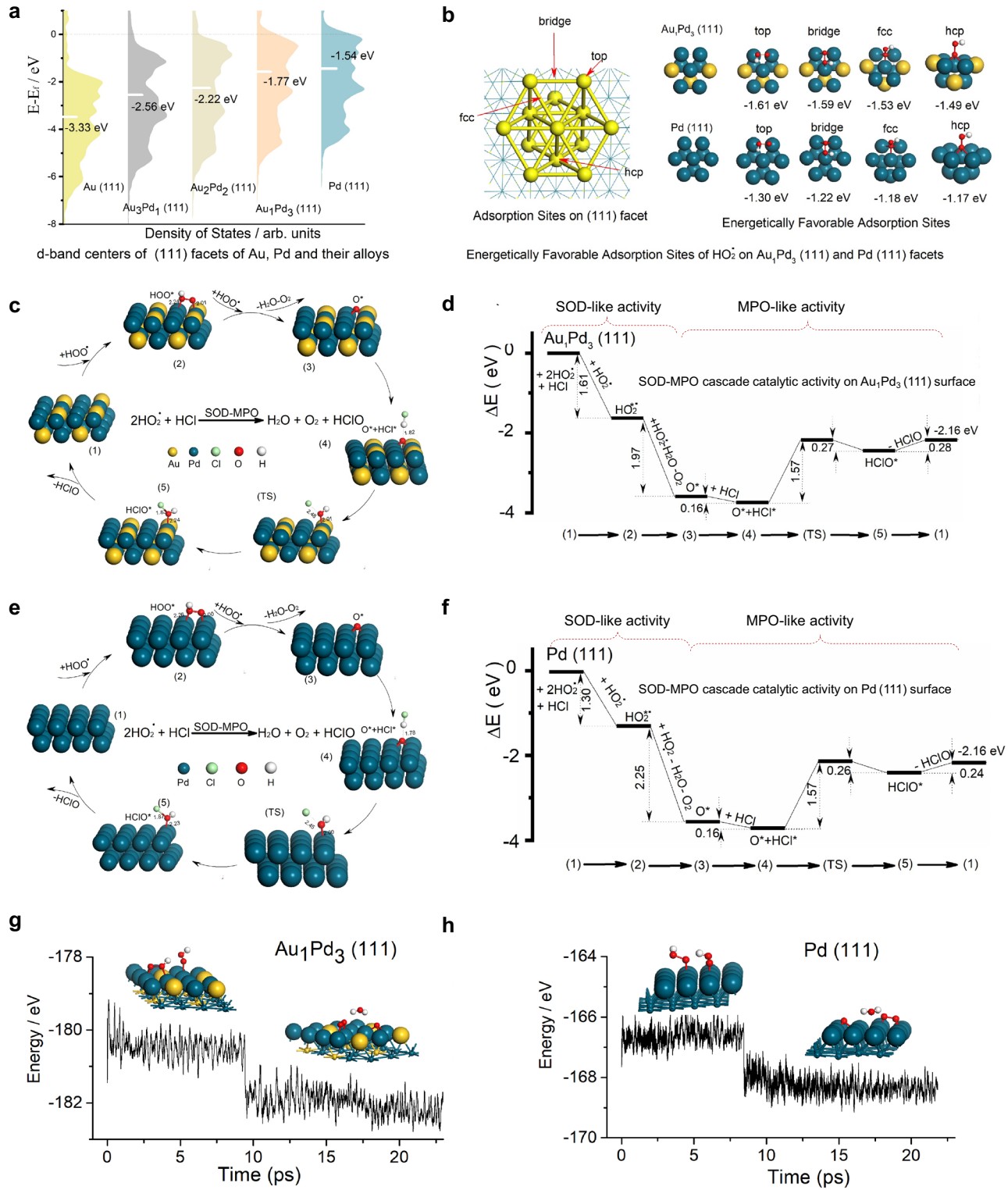

**Fig. 4 | The DFT calculations of SOD-MPO-like cascade catalytic activity on Au₁Pd₃ (111) and Pd (111) facets. a** D-band centers of (111) facets of Au, Pd, and their alloys. **b** The energetically favorable adsorption sites for HO₂· radical on Au₁Pd₃ (111) and Pd (111) facets. The proposed reaction pathway (**c**) and relative energies for key intermediate and transition-state structures (**d**) in the SOD-MPO-like cascade catalytic cycle on Au₁Pd₃ (111) surface. The proposed reaction pathway (**e**) and relative energies for key intermediate and transition-state structures (**f**) in the SOD-MPO-like cascade catalytic cycle on Pd (111) surface. **g** Free energy fluctuations with respect to time in AIMD simulations and equilibrium structures of two HO₂· radicals on Au₁Pd₃ (111) surface at 300 K. **h** Free energy fluctuations with respect to time in AIMD simulations and equilibrium structures of two HO₂· radicals on Pd (111) surface at 300 K. Energy unit: eV and bond distance unit: Å.

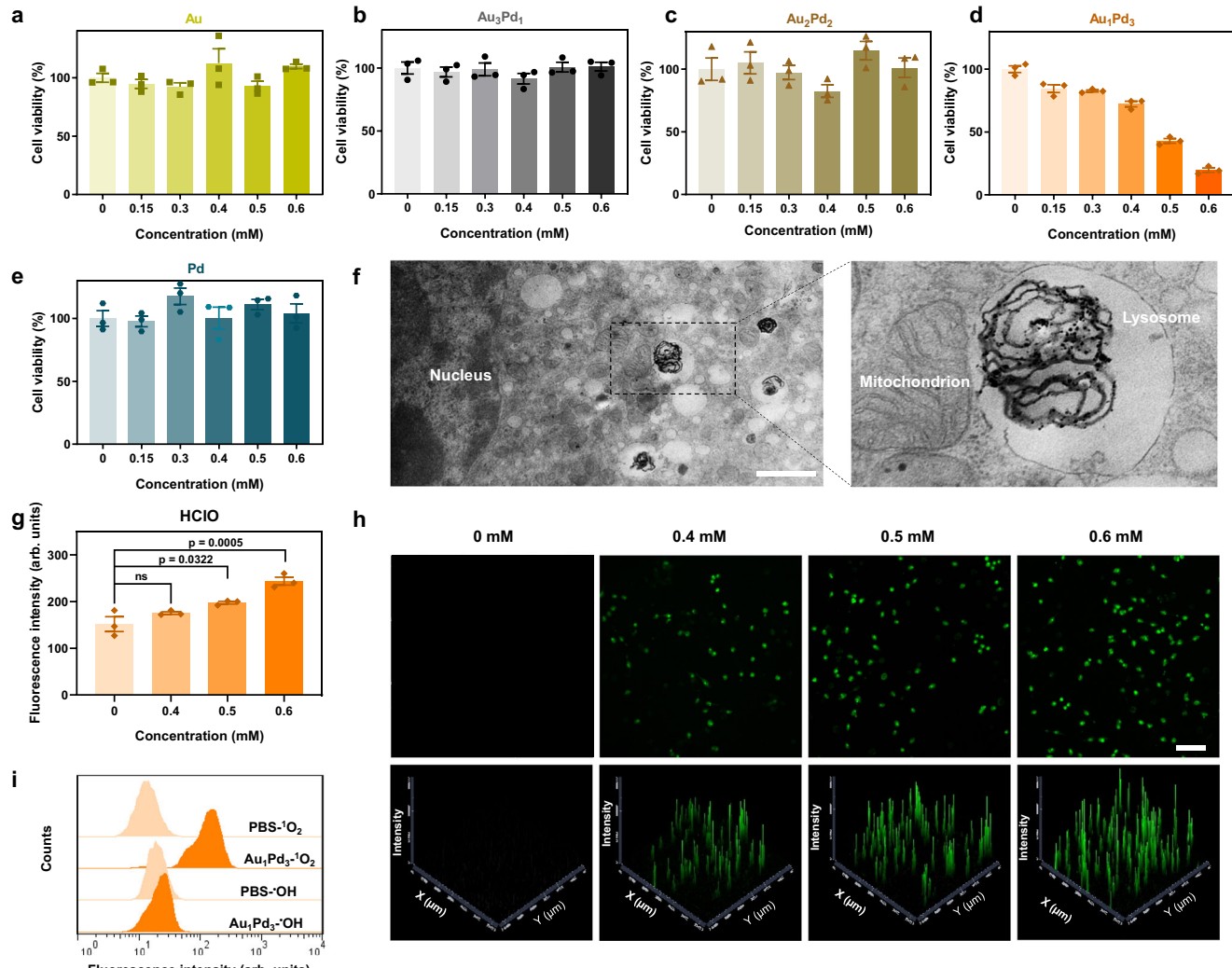

**Fig. 5 | In vitro antitumor effect of ultrasmall AuPd alloy nanozymes.** Effect of (**a**) Au, (**b**) Au$_3$Pd$_1$, (**c**) Au$_2$Pd$_2$, (**d**) Au$_1$Pd$_3$, and (**e**) Pd nanozymes on CT26 cell viability ($n = 3$ independent experiments). **f** Localization of Au$_1$Pd$_3$ nanozymes in CT26 cells. Scale bar = 1 μm. **g** HClO level in CT26 cells after exposure to different concentrations of Au$_1$Pd$_3$ nanozymes ($n = 3$ independent experiments). *P* values are determined with one-way ANOVA Tukey's multiple comparisons test. **h** Confocal microscopy images of CT26 cells after co-incubation with Au$_1$Pd$_3$ nanozymes and subsequently stained with $^1O_2$ fluorescence probe SOSG. Scale bar = 100 μm. **i** Flow cytometry analyses of CT26 cells treated with Au$_1$Pd$_3$ nanozymes using SOSG and HPF as detector for $^1O_2$ and ·OH, respectively. Representative results are present ($n = 3$ independent experiments). All data are presented as mean ± STD. Source data are provided as a Source Data file.

where $K_M^o$ represents the standard dissociation constant, R represents the gas constant, and T represents the temperature. According to the Eq. (6)[14,31], the ratio of the two $K_M^o$ values of the SOD-MPO-like cascade catalytic cycles on Pd (111) and Au$_1$Pd$_3$ (111) surfaces was about $7.3 \times 10^4$, which indicated that the SOD-MPO-like cascade catalysis was much easier to occur on Au$_1$Pd$_3$ (111) surface than on Pd (111) surface. In addition, we calculated the adsorption energy of HO$_2$· free radicals on the other three metal surfaces as presented in Supplementary Table 1. It is obvious that, for Au, Pd and their alloys, Au$_1$Pd$_3$ nanozymes showed the strongest binding energy interacting with HO$_2$· free radicals corresponding to the minimum dissociation constant and the highest SOD-MPO-like cascade catalytic activity. These computational results were in general agreement with the experimental results.

The above results demonstrated that the ratio of the two metals in AuPd alloy nanozymes significantly affect the d-band center and the adsorption energy of HO$_2$· free radicals on (111) facets of Au, Pd, and their alloys, thus influencing the catalytic performance of the nanozymes. The Au$_1$Pd$_3$ nanozymes exhibited the highest cascade activity, attributing to the high d-band center and adsorption energy for HO$_2$· free radicals. Drawing on the experimental findings and theoretical calculations, it is suggested that the ensemble effect, specifically linked to the Au: Pd ratio of nanozymes, is the key to enhancing catalysis in this study.

## In vitro antitumor effect of ultrasmall AuPd nanozymes

Given the potential of AuPd alloy nanozymes to simulate the SOD-MPO cascade reaction of neutrophils, we proceeded to test their in vitro antitumor ability. We used murine colon cancer cells CT26 as model cells. Figure 5a–e showed that among the five AuPd nanozymes tested, only Au$_1$Pd$_3$ significantly inhibited cell viability, and the inhibitory effect increased with the concentration of Au$_1$Pd$_3$ alloy nanozymes. These results were consistent with the above enzymatic activity results, where Au$_1$Pd$_3$ possessed the highest cascade activity, suggesting that the killing effects should come from the SOD-MPO-like cascade catalysis of Au$_1$Pd$_3$ alloy nanozymes. Subsequently, our investigation shifted towards studying the behavior and elucidating the killing mechanism of Au$_1$Pd$_3$ alloy nanozymes within cellular environments. Transmission electron microscopy (TEM) analyses were employed to investigate the stability of Au$_1$Pd$_3$ alloy nanozymes over a period of 72 h in both PBS and Dulbecco's Modified Eagle Medium

(DMEM). The TEM images presented in Supplementary Fig. 7 showed that Au$_1$Pd$_3$ alloy nanozymes maintained their stability without experiencing aggregation during the entire experimental duration. Furthermore, Supplementary Fig. 8 demonstrated that the size of the Au$_1$Pd$_3$ alloy nanozymes remained consistently below 6 nm throughout the observation period. These findings indicate that the Au$_1$Pd$_3$ alloy nanozymes exhibit robust stability and maintain their nanostructural integrity in both PBS and DMEM, which is crucial for their subsequent evaluation and investigation within cellular systems. Then, we investigated the subcellular localization of nanozymes in CT26 cells, as the pH and concentrations of enzyme substrates, such as Cl$^-$, vary across different intracellular compartments and environments. The TEM results showed that the Au$_1$Pd$_3$ alloy nanozymes mainly localized in the endosomes or lysosomes (Fig. 5f) surrounded by single-layer membrane and comprising tubular and multivesicular areas. These results suggest that Au$_1$Pd$_3$ nanozymes are taken up by cells through endocytosis, which is an important route internalizing nanomaterials[47], and they would ultimately be transported to lysosomes. To investigate the internalization mechanism of Au$_1$Pd$_3$ alloy nanozymes into CT26 cells, we co-incubated Au$_1$Pd$_3$ alloy nanozymes with different endocytosis inhibitors[48] such as chlorpromazine (CPZ, a clathrin-mediated endocytosis inhibitor), amiloride (AMI, a micropinocytosis-mediated endocytosis inhibitor) and genistein (GST, a caveolin-mediated endocytosis inhibitor), then detected the internalization amount of Au$_1$Pd$_3$ alloy nanozymes by the cells. FITC was modified on Au$_1$Pd$_3$ nanozymes using the same method as modifying PEG through strong coordination between metal and SH, making it feasible to quantify the internalization profile of Au$_1$Pd$_3$ using flow cytometry. The results presented in Supplementary Fig. 9 show that the uptake of Au$_1$Pd$_3$ nanozymes by cells significantly decreased in the presence of CPZ, while GST and AMI did not inhibit the internalization of them. These results indicate that Au$_1$Pd$_3$ nanozymes enter CT26 cells through clathrin-mediated endocytosis. Next, we assessed the stability of Au$_1$Pd$_3$ nanozymes in the lysosomes and found that they remain stable in a pH 4.5 lysosome-mimicking environment (Supplementary Fig. 10), which is the basis for Au$_1$Pd$_3$ nanozymes to exert their enzymatic activities in lysosomes. Worthy of attention is that the concentration of Cl$^-$ in the lysosome (80 mM) is 4-16 times higher than that in the cytoplasm (5–20 mM), as a large amount of Cl$^-$ needs to enter the low pH lysosome to maintain electrical neutrality[49]. High concentration of Cl$^-$ is beneficial for Au$_1$Pd$_3$ nanozymes to exert MPO-like activity. The product HClO from the SOD-MPO-like cascade reaction was detected by APF and HPF probes. The flow cytometry results in Fig. 5g show that Au$_1$Pd$_3$ nanozymes catalyzed the production of HClO in cells. The product $^1O_2$ can be detected using the SOSG probe[50]. Confocal microscopy images and flow cytometry results show that stimulation by Au$_1$Pd$_3$ led to the generation of a large amount of highly oxidative $^1O_2$ (Fig. 5h and Supplementary Fig. 11). Apart from $^1O_2$, the mechanism of generating ·OH through catalysis is commonly used in nanozyme tumor catalytic therapy[51]. Thus, we simultaneously detected the production of $^1O_2$ and ·OH under the stimulation of Au$_1$Pd$_3$ nanozymes, and found that while $^1O_2$ was produced in large quantities, there was no significant production of ·OH (Fig. 5i and Supplementary Fig. 12). These results indicate that the killing mechanism of Au$_1$Pd$_3$ nanozymes is different from that of traditional nanozymes.

## Tumor cell death mechanism induced by Au$_1$Pd$_3$ alloy nanozymes

Next, we sought to investigate the underlying mechanism of tumor cell death induced by Au$_1$Pd$_3$ nanozymes. To assess this, we utilized TEM to examine the morphological changes in cells following treatment with these nanozymes. As shown in Fig. 6a, distinct features indicative of cell apoptosis became evident[52]. These included the disappearance of microvilli on the cell surface, cellular shrinkage and rounding, increased cytoplasmic density, condensation of nuclear chromatin

along the nuclear membrane, and the formation of apoptotic bodies containing certain organelles. The appearance of these apoptotic markers suggests that treatment with Au$_1$Pd$_3$ nanozymes triggers apoptosis as the mode of cell death. Inspired by the condensed nuclear chromatin, we checked the DNA of cells treated with Au$_1$Pd$_3$ nanozymes to see what kind of impact they received. The generation of a large amount of phosphorylated histone H$_2$AX (γ-H$_2$AX), which is a marker of DNA double-strand breaks[53], indicates that $^1O_2$ produced by Au$_1$Pd$_3$ nanozymes cause severe DNA damage (Fig. 6b). DNA damage typically triggers the response of cells to prevent replication and transmission of damaged DNA to daughter cells[54]. To explore the effects of Au$_1$Pd$_3$ nanozymes on the cell cycle, flow cytometry analysis was conducted. The results revealed significant disruption of the cell cycle and interruption of the G$_0$/G$_1$ checkpoint transition following treatment with Au$_1$Pd$_3$ nanozymes (Fig. 6c and Supplementary Fig. 13). This disruption suggests that the damaged DNA and cell cycle abnormalities induced by the nanozymes contribute to the apoptotic process. To confirm the impact on cell viability and apoptosis, we examined cell membrane permeability and eversion of phosphatidylserine[55], both of which are indicators of cellular apoptosis. The results, presented in Fig. 6d, e, supported the occurrence of cell apoptosis following treatment with Au$_1$Pd$_3$ nanozymes. Taken together, the above results indicate that Au$_1$Pd$_3$ nanozymes cause DNA damage and cell cycle arrest through SOD-MPO-like cascade catalysis to generate $^1O_2$, leading to apoptosis of cells.

## The pharmacokinetics of ultrasmall Au$_1$Pd$_3$-FA nanozymes

Au$_1$Pd$_3$ nanozymes have exhibited good anti-tumor effects in vitro. Prior to their use in tumor therapy in vivo, we first investigated their pharmacokinetics. In order to target the tumor site in vivo, we modified them with tumor-targeting molecule folic acid (FA)[56]. Using the same method as PEG modification, we coated the nanozyme surface with SH-PEG-FA. The successful modification of FA was confirmed by FTIR and UV/Vis absorption spectra. The peak at 1607 cm$^{-1}$ in the FTIR spectrum corresponded to the bending vibration of the NH- group on FA, while the absorption peak at 285 nm in the UV−Vis absorption spectrum corresponded to the characteristic absorption peak of the pterin ring in FA[57] (Supplementary Fig. 14). Au$_1$Pd$_3$-FA nanozymes showed 5 nm ultrasmall size (Supplementary Fig. 15), indicating their potential for in vivo renal clearance ability. By detecting the production of $^1O_2$, it was confirmed that there was no significant difference in the SOD-MPO-like cascade activity between Au$_1$Pd$_3$ and Au$_1$Pd$_3$-FA (Supplementary Fig. 16). Moreover, the modification of FA did not change the internalization mechanism of Au$_1$Pd$_3$ nanozymes into CT26 cells through clathrin-mediated endocytosis (Supplementary Fig. 17). Importantly, the FA can bind to the overexpressed FA receptors on CT26 cells[14], resulting in a better cytotoxic effect of Au$_1$Pd$_3$-FA compared to Au$_1$Pd$_3$ (Supplementary Fig. 18). These results indicate that the modification of FA is beneficial for the application of Au$_1$Pd$_3$ nanozymes in the treatment of tumors in vivo.

The blood half-life of Au$_1$Pd$_3$-FA nanozymes was demonstrated to be ~2.12 h (Fig. 7a). Subsequently, we conducted an investigation to determine whether the ultrasmall (<6 nm) Au$_1$Pd$_3$-FA nanozymes possess the dual capabilities of tumor targeting and renal clearance. Au$_1$Pd$_3$-FA nanozymes were modified with Cy5.5 for in vivo visualization by the same method of PEG and FA modification mentioned above. After intravenous injection into mice, the distribution of Au$_1$Pd$_3$-FA nanozymes in various organs was assessed over time. As shown in Fig. 7b and Supplementary Fig. 19, the incorporation of FA modification facilitated the accumulation of nanozymes at the tumor site. Additionally, a substantial portion of the remaining nanozymes accumulated in the kidneys, with a gradual decrease observed over time, indicative of their progressive elimination from the body. To provide further confirmation of the elimination process, the remaining amount of Pd content in major organs was analyzed after 7 days

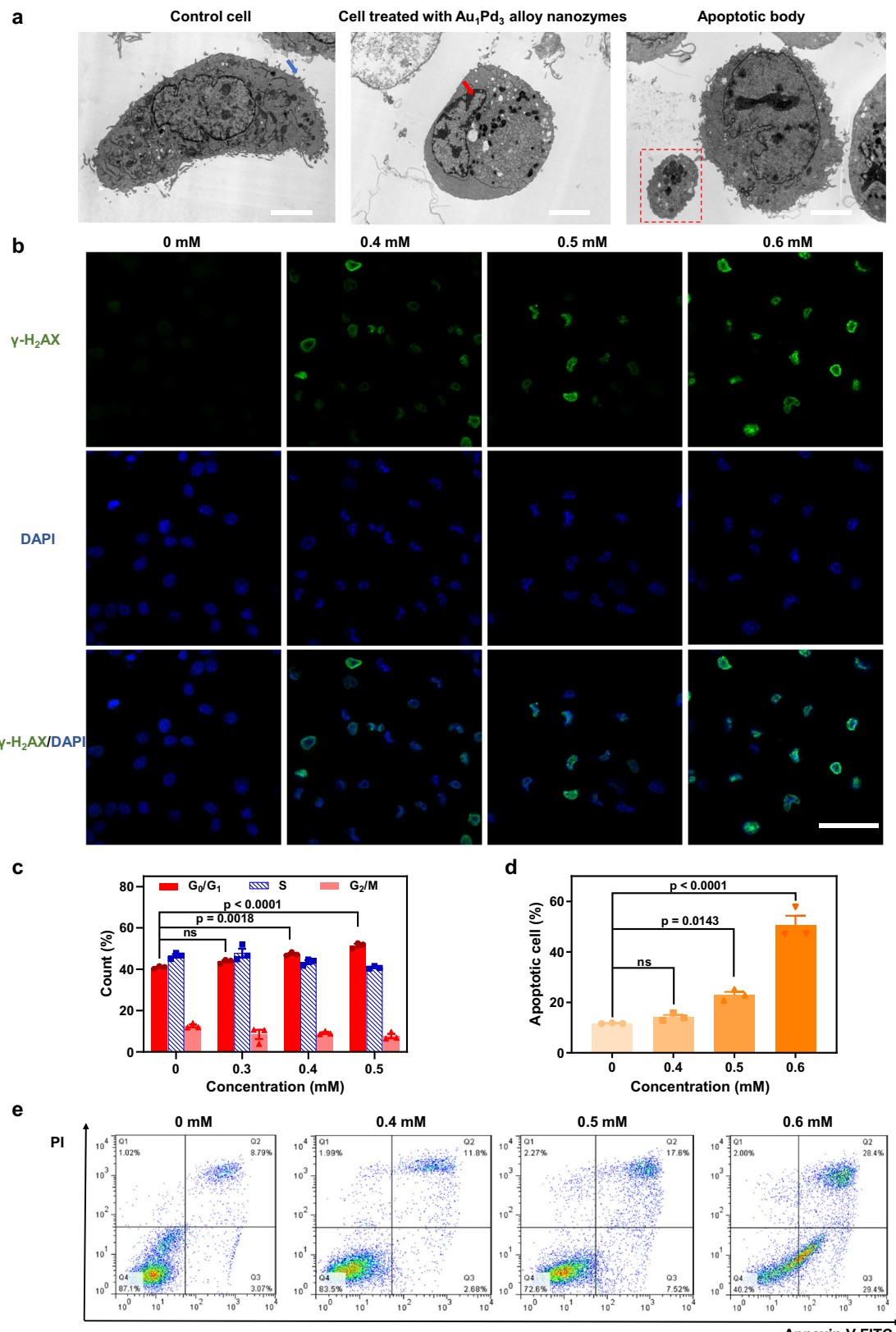

**Fig. 6 | Tumor cell death mechanism induced by Au₁Pd₃ alloy nanozymes.**
**a** Morphology of CT26 cells treated with PBS and Au₁Pd₃ nanozymes. The blue and red arrows indicate the microvilli and condensed nuclear chromatin, respectively. The red box refers to the apoptotic body. Scale bars = 4 μm. **b** Confocal microscopy images of CT26 cells treated with different concentrations of Au₁Pd₃ nanozymes using γ-H₂AX as a DNA damage biomarker. Scale bar = 50 μm. **c** Data analysis of flow cytometry assays on cell cycles of cells treated with Au₁Pd₃ nanozymes ($n = 3$

independent experiments). *P* values are determined with two-way ANOVA Tukey's multiple comparisons test. **d** Data analysis and (**e**) flow cytometry results of Annexin V-FITC/PI assay on CT26 cells treated with Au₁Pd₃ nanozymes ($n = 3$ independent experiments). *P* values are determined with one-way ANOVA Tukey's multiple comparisons test. All data are presented as mean ± STD. Source data are provided as a Source Data file.

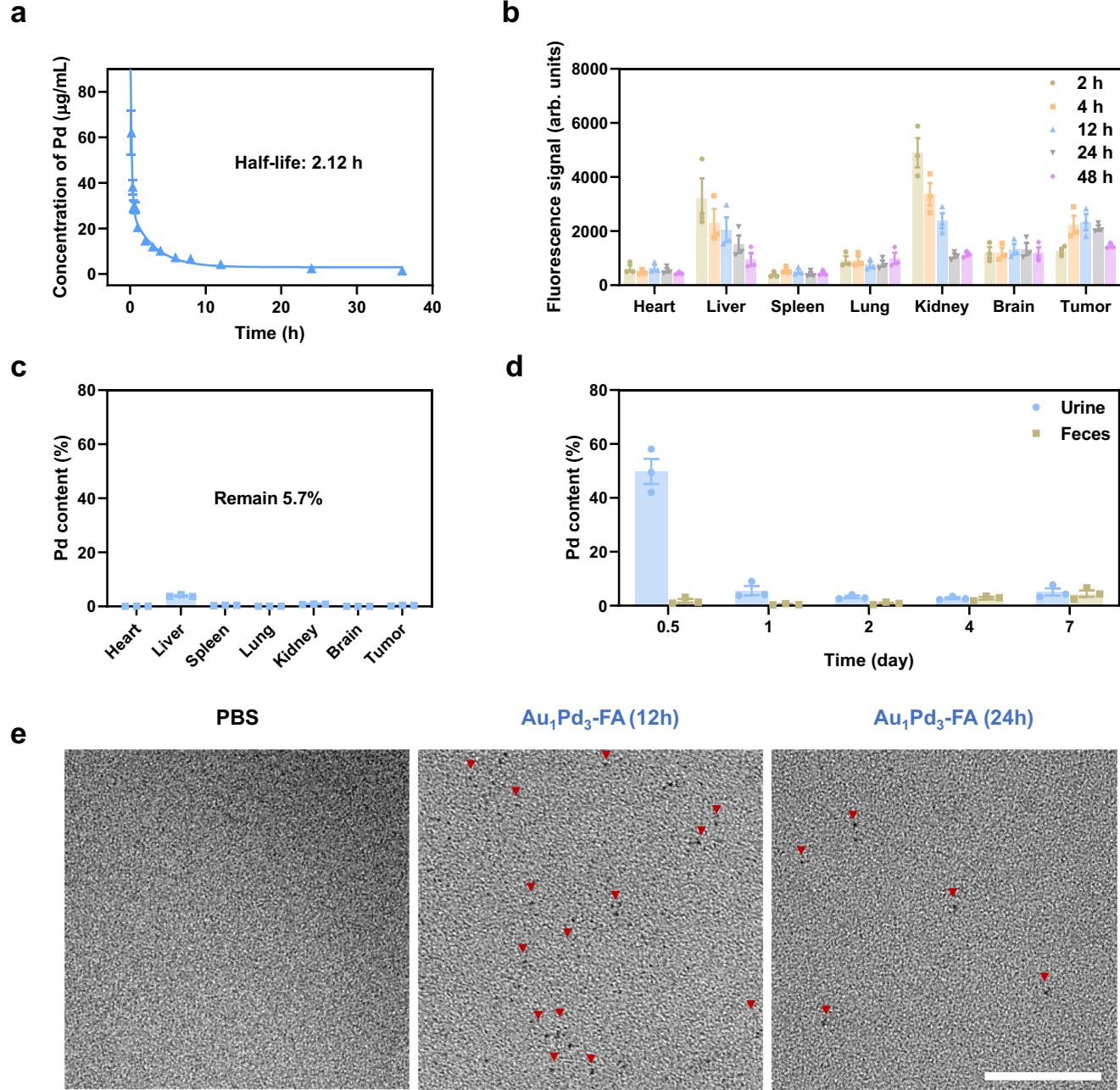

**Fig. 7 | The pharmacokinetics of ultrasmall Au₁Pd₃-FA nanozymes. a** The blood circulation curve of intravenously injected Au₁Pd₃-FA nanozymes (*n* = 4 mice). **b** The quantitative analyses of the biodistribution of Au₁Pd₃-FA-Cy5.5 nanozymes in main organs and tumors at different time points (*n* = 3 mice per group). **c** The Pd content in main organs and tumors on the seventh day after intravenous injection of Au₁Pd₃-FA-Cy5.5 nanozymes (*n* = 3 mice). **d** Pd amounts in urine and feces at various intervals from CT26 tumor-bearing mice after intravenous injection of Au₁Pd₃-FA-Cy5.5 nanozymes (*n* = 3 mice). **e** Transmission electron microscopy images of concentrated urine collected from CT26 tumor-bearing mice after intravenous injection of PBS and Au₁Pd₃-FA-Cy5.5 nanozymes. Scale bar = 50 nm. All data are presented as mean ± STD. Source data are provided as a Source Data file.

following intravenous injection of Au₁Pd₃-FA nanozymes. The results revealed a mere 5.7% retention of Pd content, indicating substantial elimination of the nanozymes from the organs within the given timeframe (Fig. 7c). Moreover, inductively coupled plasma mass spectrometer (ICP-MS) analysis was conducted to determine the Pd contents in urine and feces samples collected from mice over a period of seven days. The results demonstrated that 67% of the nanozymes were effectively cleared by the kidneys and subsequently excreted through urine. Conversely, only a negligible amount of the nanozymes was excreted through feces, as indicated in Fig. 7d. TEM images of the urine revealed the presence of intact particles (Fig. 7e), providing visual

confirmation of the kidney clearance ability of Au₁Pd₃-FA nanozymes. This efficient clearance mechanism helps prevent the prolonged accumulation of nanozymes in the body, contributing to the overall safety of their in vivo applications. Taken together, the modification with FA on Au₁Pd₃ nanozymes facilitates their enrichment at the tumor site, while the ultrasmall size of Au₁Pd₃-FA nanozymes allows them to be cleared by the kidneys and excreted from the body through urine.

## Biosafety analyses of ultrasmall Au₁Pd₃-FA nanozymes
Our Au₁Pd₃ nanozymes provide HClO and ¹O₂-elevating strategy from the endogenous O₂⁻ for tumor therapy. Actually, due to abnormal

metabolism and proliferation rate, the $O_2^-$ level in tumor sites is higher than in normal tissues[58,59], which makes $Au_1Pd_3$ nanozymes have the potential to exert tumor-specific cell killing effect. To prove the selective cytotoxicity of $Au_1Pd_3$ nanozymes on tumor cells and normal cells, we employed murine hepatocytes AML12 as normal cells and compared their response to the cytotoxic effects of $Au_1Pd_3$ and $Au_1Pd_3$-FA. Interestingly, our findings revealed that $Au_1Pd_3$ nanozymes showed concentration-dependent cytotoxic effects only on tumor cells CT26, while having no significant toxic effect on AML12 cells (Fig. 8a).

$Au_1Pd_3$-FA nanozymes exhibited enhanced specific cytotoxicity towards tumor cells (Fig. 8b) due to the high expression of FA receptors on tumor cells. Importantly, the renal clearance capability and tumor-specific cytotoxic effect of $Au_1Pd_3$-FA nanozymes would potentially have good biosafety in vivo (Fig. 8c). Thus, we conducted biosafety analyses for 7 and 28 days by intravenously injecting $Au_1Pd_3$-FA nanozymes into the bodies of healthy mice. The results showed no significant difference in body weight between mice injected with $Au_1Pd_3$-FA nanozymes and those who received PBS injection throughout the monitoring period (Fig. 8d, h). To evaluate the potential inflammatory response caused by the injection of nanozymes, we conducted blood routine tests in mice. Remarkably, our results indicated that there were no significant alterations observed in each blood parameter at both 7 days (Fig. 8e) and 28 days (Fig. 8i) following treatment with $Au_1Pd_3$-FA nanozymes. These findings provide strong evidence that the administration of $Au_1Pd_3$-FA nanozymes did not induce inflammation within the mice's bodies, highlighting the favorable biocompatibility and safety of these nanomaterials for in vivo applications. Given that these nanozymes are mainly distributed in the liver and kidneys, we tested the biomarkers of liver and kidney function alanine transaminase (ALT), aspartate transaminase (AST), alkaline phosphatase (ALP), blood urea nitrogen (BUN), and creatinine (CREA) of mice injected with $Au_1Pd_3$-FA nanozymes at 7 days and 28 days post-injection, and found no difference compared to the PBS injection group (Fig. 8f and j). Moreover, hematoxylin and eosin (HE) staining of the major organs of mice injected with $Au_1Pd_3$-FA nanozymes showed no morphological abnormality after 7 days (Fig. 8g) or 28 days (Fig. 8k). According to the above test results, it is apparent that $Au_1Pd_3$-FA nanozymes possess excellent in vivo biosafety, predominantly attributable to their ability to be cleared through the kidneys and their cytotoxicity that is specific to tumors.

### In vivo antitumor activity of $Au_1Pd_3$-FA nanozymes

Considering the excellent in vitro antitumor effects and in vivo tumor-targeting ability and biosafety of $Au_1Pd_3$-FA nanozymes, we proceeded to investigate their in vivo antitumor efficacy. Thus, we established subcutaneous CT26 tumor-bearing mice and administered intravenous injections of PBS, $Au_1Pd_3$ and $Au_1Pd_3$-FA nanozymes (Fig. 9a). Analysis of the tumor growth curve revealed that both $Au_1Pd_3$ and $Au_1Pd_3$-FA nanozymes effectively inhibited tumor growth compared to the PBS injection group. Notably, the inhibitory effect of $Au_1Pd_3$-FA nanozymes was even more pronounced (Fig. 9b). In fact, complete tumor regression was observed in one of the mice, highlighting the potent tumor-targeting ability of $Au_1Pd_3$-FA nanozymes (Fig. 9c). The harvested tumor tissues were weighed, showing a similar trend to the volume curve of the tumors (Fig. 9d). Importantly, there was no significant impact on the body weight of mice from $Au_1Pd_3$ and $Au_1Pd_3$-FA nanozymes-treated groups compared to the PBS group, indicating the safety of the treatment (Fig. 9e). The excellent tumor inhibitory effect and good safety profile greatly resulted in a significant improvement in the survival rate of mice in the nanozyme treatment groups (Fig. 9f).

Futhermore, we harvested the tumor tissue to examine the tissue-level antitumor effect and mechanism. The results showed that both $Au_1Pd_3$ and $Au_1Pd_3$-FA nanozymes induced the generation of HClO and

$^1O_2$, leading to DNA damage. Notably, $Au_1Pd_3$-FA nanozymes exhibited a more pronounced effect (Fig. 9g). The induction of tumor cell apoptosis was confirmed through TdT-mediated dUTP nick-end labeling (TUNEL) assays. Histological examination of tumor tissue using HE staining revealed reduced tumor cell density, presence of gaps, and deep nuclear staining, particularly in the groups treated with nanozymes, especially $Au_1Pd_3$-FA nanozymes. Taken together, these results indicate that $Au_1Pd_3$ nanozymes exert SOD-MPO-like cascade activity in vivo to produce HClO and $^1O_2$, leading to apoptosis of tumor tissue. The modification with FA enhances the tumor-targeting ability of the nanozymes, resulting in improved therapeutic efficacy in CT26 tumors with high expression of FA receptors.

To further validate these findings, we constructed another tumor xenograft model using 4T1 mammary cancer cells, which are also known to exhibit high expression of FA receptors[60]. The in vivo antitumor efficacy of nanozymes was further verified in this model (Fig. 9h). The results demonstrated that the growth of 4T1 tumors was significantly inhibited in the $Au_1Pd_3$ nanozyme-treated group, with an even greater inhibition observed in the $Au_1Pd_3$-FA nanozyme-treated group (Fig. 9i). Consistent with the tumor growth inhibition, the tumor weight results showed a similar trend (Fig. 9j). In addition, the body weight of the mice treated with $Au_1Pd_3$ and $Au_1Pd_3$-FA nanozymes was not significantly affected, proving the safety of treatment (Fig. 9k). The excellent tumor suppression effect and favorable safety profile significantly prolonged the survival rate of mice in the $Au_1Pd_3$ and $Au_1Pd_3$-FA nanozymes-treated groups (Fig. 9l). These results demonstrate the potential of $Au_1Pd_3$ nanozymes, which mimic the neutrophil SOD-MPO cascade killing mechanism, as a promising therapeutic agent for various types of tumors.

## Discussion

Enzymes that have undergone long-term evolutionary processes play an important role in the immune-mediated killing performed by natural immune cells like neutrophils. These enzymes provide valuable guidance and templates for the development of innovative cancer treatment strategies. Neutrophils efficiently execute their killing function through a cascade catalytic reaction of SOD and MPO, which leads to the production of highly oxidizing species like HClO and $^1O_2$. In this work, we synthesized five types of AuPd alloy nanozymes: Au, $Au_3Pd_1$, $Au_2Pd_2$, $Au_1Pd_3$, and Pd. Remarkably, all of these nanozymes exhibited SOD-like and MPO-like activities, with their catalytic performance closely associated with the alloy ratio. Among them, $Au_1Pd_3$ showed the highest cascade enzymatic activity, enabling it to simulate the killing mechanism of neutrophils for in vitro and in vivo tumor treatment. Therefore, we have developed a biomimetic tumor catalytic therapy based on the SOD-MPO-like cascade activity of nanozymes. The key advantage of our MPO-like system lies in its reliance on nanoparticle-based inherent catalytic activity without the need for natural enzymes. This difference allows for several notable advantages. Firstly, due to the dual enzymatic activities of the AuPd alloy nanozyme, our system only requires one type of nanozyme to achieve the SOD-MPO cascade reaction, making the entire system relatively simple. In contrast, the natural enzyme-based system requires two types of natural enzymes and enzyme carrier. Secondly, the use of nanomaterials in our system provides improved stability compared to natural enzymes. Nanoparticles are known for their robustness and resistance to degradation, rendering our system more durable and suitable for long-term applications. Furthermore, the synthesis of nanoparticles can be achieved with relative ease and on a large scale compared to the production of natural enzymes, which often requires complex and time-consuming purification processes. Lastly, a nanozyme-based system can achieve precise control and optimization of catalytic performance through structural regulation. However, the manipulability of natural enzyme-based system in terms of activity regulation is relatively low. These advantages underscore the potential of our

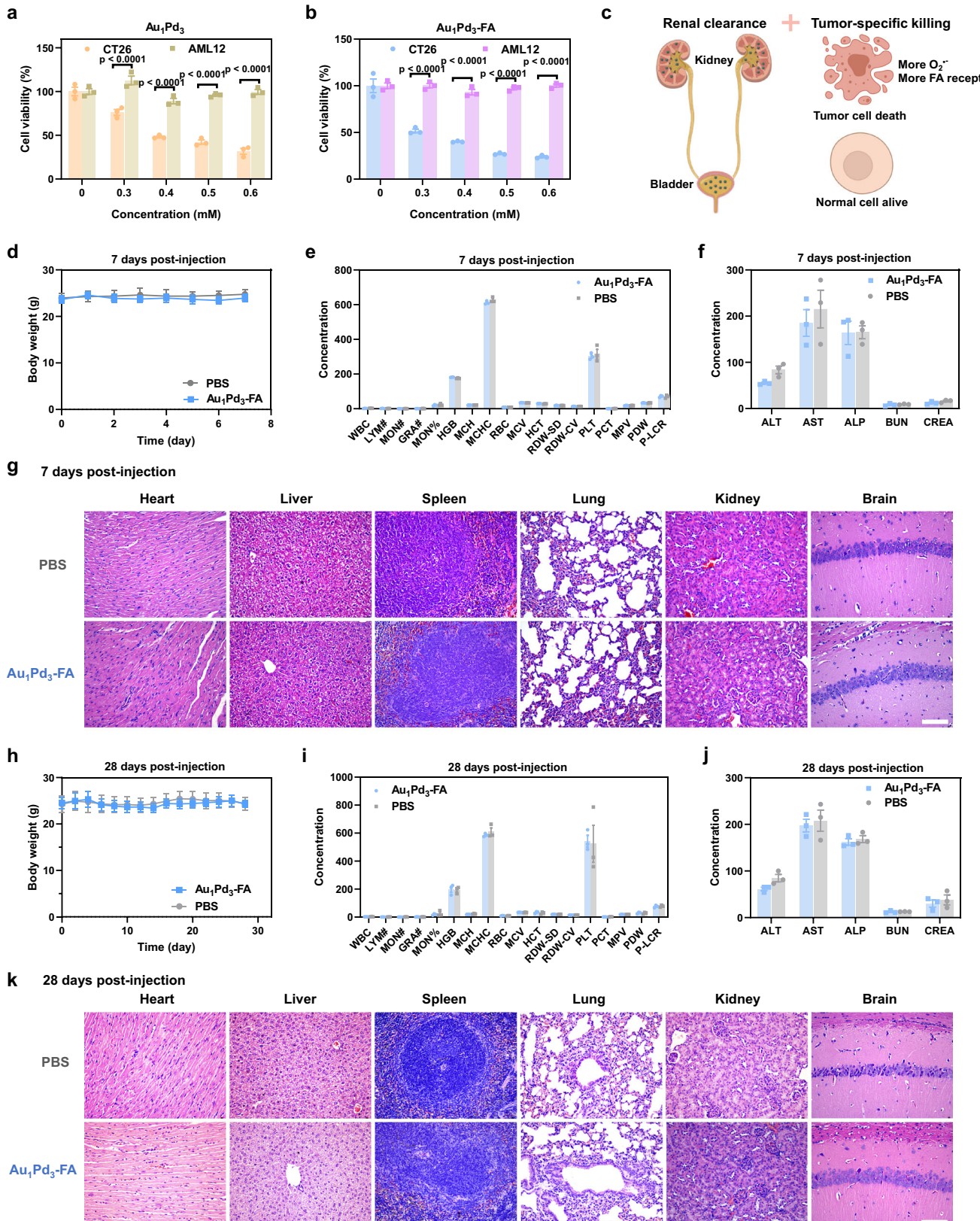

MPO-like system as a promising alternative and advance in the field of catalysis and biomedical applications.

Importantly, the use of ultrasmall AuPd alloy nanozymes in this tumor treatment approach offers excellent biosafety due to their ability to undergo renal clearance and their tumor-specific killing effect. The effectiveness, safety, and simplicity of this treatment, which does not necessitate additional therapeutic interventions, make it highly promising for clinical translation. In addition, we employed theoretical calculations to elucidate the catalytic mechanism underlying the SOD-like and MPO-like activities of AuPd nanozymes and their relationship with the alloy ratio. This provides some general guidelines for the development of nanozymes as follows. A high d-band center for

Fig. 8 | Biosafety analyses of ultrasmall Au₁Pd₃-FA nanozymes. Effect of (a) Au₁Pd₃ nanozymes and (b) Au₁Pd₃-FA nanozymes on the cell viability of CT26 and AML12 cells (*n* = 3 independent experiments). *P* values are determined with two-way ANOVA Sidak's multiple comparisons test. **c** Illustration of the renal clearance ability and tumor-specific killing effect. Graphical content was created with BioRender.com. **d** Body weight changes in mice 7 days and **h** 28 days after the treatment with Au₁Pd₃-FA nanozymes and PBS, respectively (*n* = 3 mice per group). **e** Routine blood tests of mice at 7 days and (**i**) 28 days post-injection of Au₁Pd₃-FA nanozymes and PBS, respectively (*n* = 3 mice per group). **f** Biomarkers of liver and kidney function analyses of mice at 7 days and (**j**) 28 days post-injection of Au₁Pd₃-FA nanozymes and PBS, respectively (*n* = 3 mice per group). **g** HE staining images of the tissue sections of heart, liver, spleen, lung, kidney, and brain obtained from mice after treatment with Au₁Pd₃-FA nanozymes and PBS at 7 days and (**k**) 28 days post-injection. Scale bar = 100 μm. All data are presented as mean ± STD. Source data are provided as a Source Data file.

metal surface and high adsorption energy of metal surface for enzymatic substrates contribute to a higher catalytic activity. Based on the above two factors, alloy nanozymes generally exhibit higher activity compared to monometallic nanozymes. However, the specific alloy ratio for optimal enzymatic activity depends on the specific metal and enzymatic activity. This work will also promote the discovery and design of nanozymes with MPO-like activity, which has been less explored thus far. Furthermore, the development of biomimetic tumor therapy described in this study is expected to catalyze the advancement of further biomimetic treatments for tumors and other diseases.

Indeed, the study also has a few limitations, such as the presence of by-products in the synthetic process of ultrasmall AuPd alloy nanozymes. It is crucial to ensure that the studied AuPd alloy nanozymes are of ultrasmall size, as the ultrasmall dimension of alloy nanozymes contributes to their high enzymatic activities and their ability to be cleared by the kidneys. However, some larger particles are formed as byproducts, thus necessitating the use of ultrafiltration tubes to eliminate these particles and guarantee the uniformity of the ultrasmall nanozymes (described in the method section). Consequently, it would be worthwhile to further optimize the method employed for ultrasmall nanozyme synthesis, with the aim of simplifying the process and enhancing the overall yield. Another challenge in the research is how to achieve the renal clearance ability and tumor targeting ability of the ultrasmall Au₁Pd₃ nanozymes simultaneously. The ultrasmall size of the nanozymes enables them rapid metabolism by the kidneys. However, this also leads to limited accumulation of the nanozymes at the tumor site. To address this issue, we introduced a tumor-targeting molecule, folic acid (FA), to modify the ultrasmall Au₁Pd₃ nanozymes. Despite this modification, half of the nanozymes were still eliminated from the body through urine, as depicted in Fig. 7d. Therefore, further investigation is necessary to strike a balance between the renal clearance and tumor targeting abilities. For instance, optimizing the modification density of tumor-targeting molecules or exploring alternative tumor-targeting molecules are both worth considering. Regarding the alloy effect in determining the catalytic activity of ultrasmall AuPd alloy nanozymes, our experimental and theoretical calculation results demonstrated that the ensemble effect that is related to the ratio of the two metals in AuPd alloy nanozymes significantly affect the catalytic performance of the nanozymes. However, the ligand and strain effects may also have influence on the electronic structure, bond lengths of the AuPd alloys, thus affecting their catalytic activity. The more in-depth study about the alloy effect in ultrasmall AuPd alloy nanozymes will be the subject of our future research.

## Methods
### Ethical regulations
All research complied with the relevant ethical regulations. The animal studies were conducted in accordance with the approved protocol of the Institutional Animal Care and Use Committee (IACUC) of the Institute of Biophysics, Chinese Academy of Sciences (Project number: SYXK2023168). 6–8-week-old female BALB/c mice were purchased from Spiff (Beijing) Biotechnology Co., Ltd. All mice were group-housed 5 mice per cage in a specific pathogen-free environment in temperature (22–26 °C) and humidity (40–70%) house rooms on a 12 h light, 12 h dark cycle. The IACUC permits a maximum tumor size of 15 mm in diameter, in our work no mice exceeded this criterion.

According to the IACUC guidelines, mice that experience weight loss exceeding 20% or display symptoms such as hunched posture, impaired locomotion, or respiratory distress should be promptly euthanized using CO₂ gas. Otherwise, the mice were euthanized at the conclusion of the experiment.

### Chemicals and materials
THPC was purchased from Aladdin Chemistry Co., Ltd (Shanghai, China). HAuCl₄ and Na₂PdCl₄ were purchased from InnoChem Science & Technology Co., Ltd (Beijing, China). SH-PEG-OMe (Mw: 5000 Da), SH-PEG-FA (Mw: 5000 Da) and SH-PEG-Cy5.5 (Mw: 5000 Da) were purchased from Shanghai Tuoyang Biotechnology Co., Ltd. SOD assay kit, Cell Counting Kit-8 (CCK-8), Cell Cycle Assay Kit - PI/RNase Staining and Annexin V, FITC Apoptosis Detection Kit were purchased from Dojindo Laboratories (Kumamoto, Japan). MCD was purchased from Alfa Aesar. Pyrogallol was purchased from Energy Chemical (Shanghai, China). ABDA was purchased from Sigma-Aldrich. APF and HPF probes were purchased from AAT Bioquest Inc. SOSG probe was purchased from Thermo Fisher Scientific Inc. DNA Damage Assay Kit by γ-H2AX Immuno fluorescence and One Step TUNEL Apoptosis Assay Kit were purchased from Beyotime Biotechnology Co., Ltd (Shanghai, China).

### Instrumentation
HRTEM images were obtained using a FEI Tecnai G2 F30 transmission electron microscope, and particle sizes were calculated using Image Pro Plus software. The FTIR spectra were collected using a Nicolet™ iS™ 10 FTIR spectrometer. The Raman spectra were obtained using a Renishaw inVia Qontor instrument with 532 nm excitation. ICP-OES measurements were collected using an Agilent ICP-OES 730 instrument. ICP-MS measurements were collected using an Agilent ICP-MS 7800 instrument. XRD patterns were obtained using a Bruker D8 Advance X-ray powder diffractometer with Cu Kα radiation. XPS experiments were performed using a Thermo escalab 250Xi spectrometer. DLS experiments were carried out using a Wyatt DynaPro NanoStar 271-DPN dynamic light scattering equipment. TEM images were obtained with a FEI Tecnai Spiritelectron microscope operated at 120 kV. Flow cytometry data was collected using the FACS CaliburTM instrument from Becton Dickinson (USA). Confocal images were obtained using a ZIESS-LSM700 (Germany), the images were analyzed by ZEN 2012 software. Fluorescence imaging was carried out using an in vivo imaging instrument (IVIS Lumina 3, PE, USA), and the images were analyzed through the IVIS Living Image 3.0 software (PerkinElmer, USA).

### Synthesis of ultrasmall metal alloy nanozymes
The ultrasmall metal alloy nanozymes were synthesized using a modified version of a previously reported method[19]. To a solution of 100 mL water, THPC (4 μL) and NaOH (2 M, 667 μL) were added and stirred for 5 min in an ice-water bath. Next, metal precursors (with a total amount of 57 μmol HAuCl₄ and/or Na₂PdCl₄) were added to the mixed solution and stirred for 10 min. Subsequently, SH-PEG-OMe or SH-PEG-FA (11 mg mL⁻¹, 1 mL) was added to the solution and stirred at 30 °C for 1.5 h. To wash the nanozymes and remove large nanoparticles, the solution was subjected to ultrafiltration using a tube (Mw cutoff: 10,000 Da and 100,000 Da). The concentrations of Au and Pd in the AuPd alloy nanozymes were measured by ICP-OES. The specific

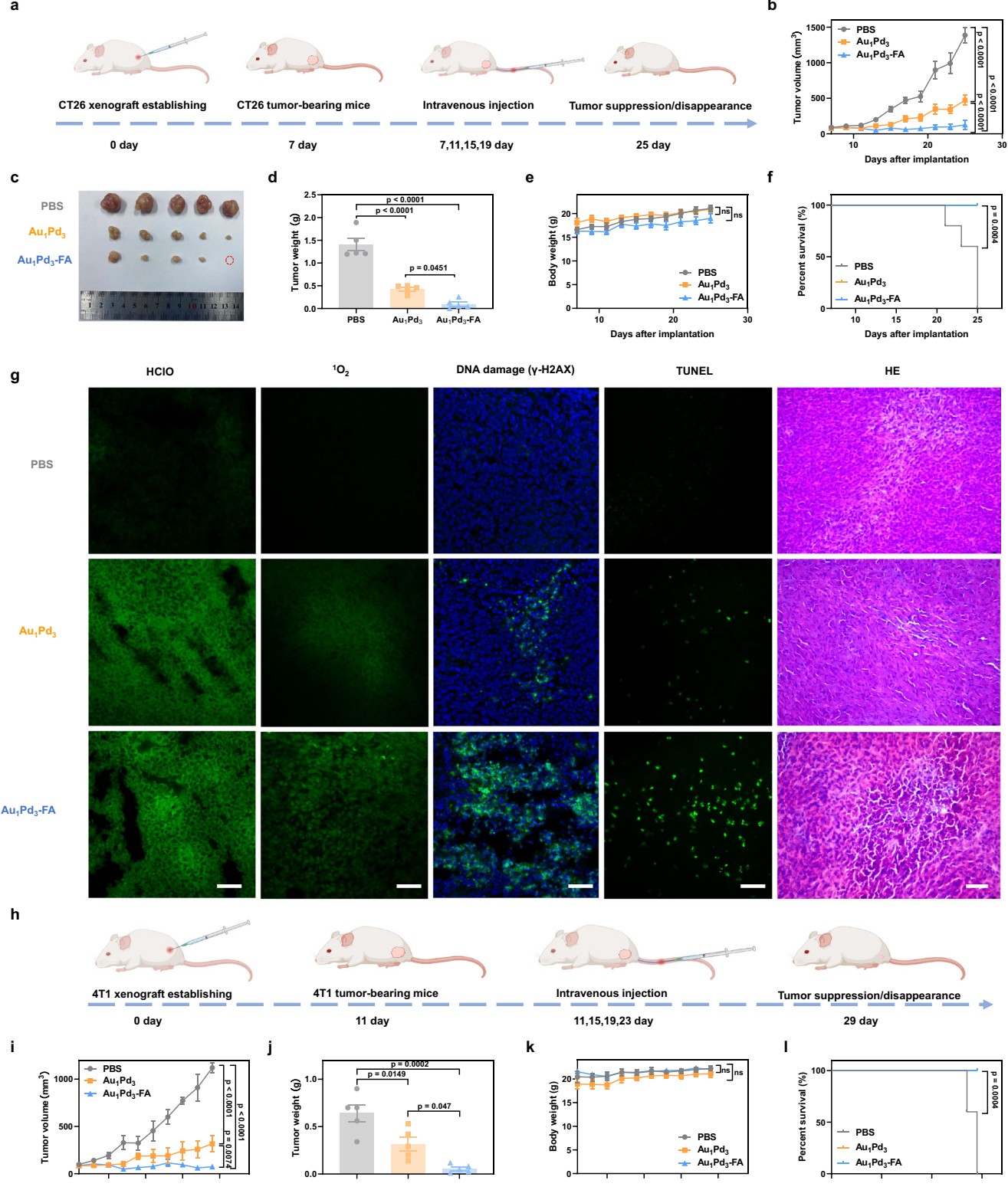

protocol of the sample preparation is as follows: the sample was placed in a sample tube, 0.5 mL of aqua regia was added, and then the sample tube was heated at 120 °C for about 30 min to dissolve. After the sample was cooled to room temperature, it was diluted to 5 mL with ultrapure water for measurement.

## SOD-like activity assay

The SOD-like activity of ultrasmall metal alloy nanozymes was assayed using a SOD assay kit (WST-1). Various concentrations of nanozymes and substrate were incubated, and the absorbance changes of WST-1 at 450 nm were collected to calculate the SOD-like activity based on the inhibition rate.

## MPO-like activity assay

The MPO-like activity of ultrasmall metal alloy nanozymes was assayed in acetate buffer (0.1 M, pH 4.5). A mixture of nanozymes (5 mM, 10 µL), KCl (1.6 M, 10 µL), $H_2O_2$ (40 mM, 10 µL), and MCD (2 mM, 10 µL) was combined with acetate buffer to a total volume of 200 µL. The

**Fig. 9 | In vivo tumor catalytic therapy of Au₁Pd₃-FA nanozymes against CT26 and 4T1 xenograft. a** Schematic illustration of CT26 tumor xenograft establishment, nanozymes administration modalities, and therapeutic outcome. **b** Tumor volume after Au₁Pd₃ and Au₁Pd₃-FA nanozymes and PBS treatment ($n = 5$ mice per group). $P$ values (at endpoint) are determined with two-way ANOVA Tukey's multiple comparisons test. **c** Photos of the dissected tumors from mice at the end of treatments (day 25). **d** Weight of the dissected tumors at the end of treatments ($n = 5$ tumors per group). $P$ values are determined with one-way ANOVA Tukey's multiple comparisons test. **e** Body weight changes of mice treated with Au₁Pd₃ and Au₁Pd₃-FA nanozymes and PBS ($n = 5$ mice per group). $P$ values (at endpoint) are determined with two-way ANOVA Tukey's multiple comparisons test. **f** Survival percentages of each group mice ($n = 5$ mice per group). $P$ values are determined with log-rank Mantel-Cox test. **g** Histopathology analyses of the tumor tissue after the treatment of Au₁Pd₃ and Au₁Pd₃-FA nanozymes and PBS: HClO, ¹O₂, DNA damage, TUNEL, and HE staining, respectively. For confocal images, scale bar = 50 μm, for HE images, scale bar = 100 μm. **h** Schematic illustration of 4T1 tumor xenograft establishment, nanozymes administration modalities, and therapeutic outcome. **i** Tumor volume after Au₁Pd₃ and Au₁Pd₃-FA nanozymes and PBS treatment ($n = 5$ mice per group). $P$ values (at endpoint) are determined with two-way ANOVA Tukey's multiple comparisons test. **j** Weight of the dissected tumors at the end of treatments ($n = 5$ tumors per group). $P$ values are determined with one-way ANOVA Tukey's multiple comparisons test. **k** Body weight changes of mice treated with Au₁Pd₃ and Au₁Pd₃-FA nanozymes and PBS ($n = 5$ mice per group). $P$ values (at endpoint) are determined with two-way ANOVA Tukey's multiple comparisons test. **l** Survival percentages of each group mice ($n = 5$ mice per group). $P$ values are determined with log-rank Mantel-Cox test. All data are presented as mean ± STD. Source data are provided as a Source Data file.

solution was incubated at 37 °C for 30 min, and the absorbance change of MCD at 290 nm was measured using a microplate reader.

## Cascade catalytic activity assay

The cascade catalytic activity of ultrasmall metal alloy nanozymes was assayed in Tris-HCl buffer (50 mM, pH 8.0). A mixture of nanozymes (2.5 mM, 10 μL), KCl (1.6 M, 10 μL), pyrogallol (0.6 M, 10 μL), and APF (1 mM, 2 μL) or HPF (1 mM, 2 μL) or ABDA (2 mM, 10 μL) was combined with Tris-HCl buffer to a total volume of 200 μL. The solution was incubated at 37 °C for 15 min. The fluorescence change of APF at 515 nm with excitation at 490 nm was measured using fluorescence spectrophotometer and the absorbance change of ABDA at 378 nm was measured using a microplate reader.

## Cell viability assay

CT26 cell line was purchased from ATCC, CRL-2638. AML12 cell line was purchased from Hunan Fenghui Biotechnology Co., Ltd, CL0550. Tumor cells CT26 or normal cells AML12 were plated in 96-well plates at a cell density of $10^4$ cells per well. They were then allowed to adhere overnight. Various concentrations of ultrasmall metal alloy nanozymes were subsequently added to the wells for a 24 h incubation period. The cell viability was then determined using a CCK-8 kit, following the manufacturer's instructions. Finally, the absorbance at 450 nm was measured using a microplate reader.

## TEM images of CT26 cells treated with ultrasmall Au₁Pd₃ nanozymes

The localization of ultrasmall Au₁Pd₃ nanozymes in cells and the effects of treatment with Au₁Pd₃ nanozymes on cell morphology were studied using TEM. CT26 cells were seeded at a density of $10^6$ cells per dish in 100 mm culture dishes and were allowed to adhere and grow overnight. The culture medium was then supplemented with Au₁Pd₃ nanozymes (0.4 mM), followed by a 24 h incubation. After incubation, the medium was removed, and the cells were detached and rinsed with a PBS buffer. They were then fixed overnight at 4 °C using 2.5% glutaraldehyde in 100 mM PBS. The cells were then postfixed in 1% (wt/vol) osmium tetraoxide in PBS, followed by dehydration using ethanol, and ultimately embedded in resin. The ultrathin sections (70 nm) were examined.

## Cellular HClO and ·OH assay

The cellular assay for HClO and ·OH was conducted using APF and HPF probes via flow cytometry. Initially, CT26 cells were plated in six-well plates at a density of $1.5 \times 10^5$ cells per well and allowed to adhere overnight. Various concentrations (0, 0.4, 0.5, 0.6 mM) of Au₁Pd₃ nanozymes were then added to the wells for 24 h of incubation. Afterward, the medium was removed, and the cells were detached and washed with a PBS buffer. Next, the cells were treated with APF or HPF probe (10 μM) at 37 °C for 30 min, followed by another PBS buffer rinse. The fluorescence emission spectrum of APF and HPF (Ex/Em = 490/515 nm) was immediately measured using flow cytometry, and the results were analyzed using FlowJo 7.6 software. While the APF probe reacts with both HClO and ·OH, the HPF probe only reacts with ·OH. By utilizing both probes, HClO can be detected.

## Cellular ¹O₂ assay

The cellular assay for ¹O₂ was conducted using SOSG probe via flow cytometry and confocal laser scanning microscopy. Initially, CT26 cells were plated in six-well plates or glass bottom cell culture dish at a density of $1.5 \times 10^5$ cells per well and allowed to adhere overnight. The cell processing method used prior to flow cytometry detection is identical to the method employed before detecting intracellular HClO, except for the use of SOSG (5 μM) as a probe. The fluorescence emission spectrum of SOSG (Ex/Em = 504/525 nm) was immediately measured using flow cytometry, and the results were analyzed using FlowJo 7.6 software. To observe intracellular ¹O₂ using confocal laser scanning microscopy, the cells do not need to be digested. They should be washed with PBS, incubated with SOSG probe, washed again with PBS, and then imaged.

## DNA damage/cell cycle/apoptosis detection

These assays were conducted using SOSG probe via flow cytometry and confocal laser scanning microscopy. Initially, CT26 cells were plated in six-well plates or glass bottom cell culture dish at a density of $1.5 \times 10^5$ cells per well and allowed to adhere overnight. Different concentrations (0, 0.4, 0.5, 0.6 mM) of Au₁Pd₃ nanozymes were then added to the wells for 24 h of incubation. The cell processing and analysis were carried out in accordance with the manufacturer's instructions. The cells were detected for DNA damage using a confocal laser scanning microscope and the cell cycle and apoptosis status were detected using flow cytometry.

## In vivo pharmacokinetics of ultrasmall Au₁Pd₃-FA nanozymes

To determine the blood circulating half-life of ultrasmall Au₁Pd₃-FA nanozymes, we intravenously injected them into four healthy female mice. At specific time intervals (0.167, 0.333, 0.5, 0.667, 1, 2, 3, 4, 6, 8, 12, 24, 36 h), we collected 10 μL of blood by puncturing the tail vein and dispersed it into aqua regia to dissolve any remaining ultrasmall Au₁Pd₃-FA nanozymes. We measured the concentration of Pd ions using ICP-MS.

In order to assess the renal clearance ability and tumor site accumulation of Au₁Pd₃-FA nanozymes, we modified the Au₁Pd₃-FA nanozymes with Cy5.5 by using the same method of PEG and FA modification as mentioned above, for in vivo visualization. Female CT26 tumor-bearing mice were obtained by subcutaneously injecting $2 \times 10^6$ cells in 100 μL of PBS buffer. Once the tumor volume reached 200–300 mm³, we injected the mice with Cy5.5- Au₁Pd₃-FA nanozymes intravenously ($n = 3$ per group). At 2, 4, 12, 24, 48 h after injection, we excised major organs, including the heart, liver, spleen, lung, kidney, brain, and tumor. We performed fluorescence imaging to determine the accumulation of the nanozymes in these organs.

We collected urine and feces from mice at various times (0.5, 1, 2, 4, 7 days) within a 7-day period. The contents of Pd were determined by ICP-MS to evaluate the excretion of $Au_1Pd_3$-FA nanozymes through urine and feces. Urine is also observed under a TEM to detect the presence of $Au_1Pd_3$-FA nanozymes. Additionally, we collected major organs at the end of the 7 days to test their Pd content and determine the remaining amount of $Au_1Pd_3$-FA nanozymes in the mice.

## Biosafety assay of $Au_1Pd_3$-FA nanozymes

For the biosafety assay of $Au_1Pd_3$-FA nanozymes, healthy female mice were randomly divided into four groups, with three mice per group. Four groups were studied as follows: mice were injected with $Au_1Pd_3$-FA nanozymes (0.3 mmol $kg^{-1}$) or PBS every two days for a total of four injections and observed for 7 days, mice were injected with $Au_1Pd_3$-FA nanozymes (0.3 mmol $kg^{-1}$) or PBS every 2 days for a total of four injections and observed for 28 days. The body weights of mice were monitored to assess their physical condition. The mice were euthanized at the designated time, and blood samples were collected for standard hematology and serum biochemistry tests at Beijing Medical Discovery Leader Biotech Co., Ltd. Additionally, the primary organs (heart, liver, spleen, lung, and kidney) were fixed in 4% paraformaldehyde, embedded in paraffin, sliced to -5 μm, and subjected to HE staining to assess their tissue morphology.

## In vivo antitumor activity of $Au_1Pd_3$-FA nanozymes

The female CT26 tumor-bearing mice were utilized for anti-tumor detection once their tumor volume reached 50–100 $mm^3$. The following equation was used to calculate the tumor volume: volume = 0.5 × (tumor length) × (tumor width)$^2$. Fifteen female mice bearing CT26 tumors were randomly divided into three groups consisting of five mice in each group. They were then intravenously injected with $Au_1Pd_3$ nanozymes (0.3 mmol $kg^{-1}$), $Au_1Pd_3$-FA nanozymes (0.3 mmol $kg^{-1}$), or PBS every 2 days for four times. The measurements of tumor volumes and body weights were taken every 2 days throughout the treatment. All mice were sacrificed when the PBS-treated mice reached an average tumor volume of 1000 $mm^3$. The tumor tissues were then harvested, weighed, and photographed. For HClO and $^1O_2$ detection, the tumors were cryosectioned at a thickness of 6 μm and stained with APF and SOSG probes. For DNA damage, TUNEL apoptosis, and HE assays, the tumors were conventionally paraffin-embedded, sectioned at a thickness of 4 μm, and placed on slides. Commercial DNA damage and TUNEL apoptosis kits were used according to the manufacturers' instructions for the detections. A standard procedure was used for the HE assays for histopathological evaluation. The survival of the mice was monitored every 2 days throughout the study. Upon reaching a tumor size of 1000 $mm^3$, the mice were euthanized and the survival curves were generated based on the remaining number of mice. The in vivo anti-tumor detection on mice bearing 4T1 tumors using $Au_1Pd_3$-FA followed the same method as mentioned above for mice bearing CT26 tumors. 4T1 cell line was purchased from ATCC, CRL-2539.

## Statistical analysis

General statistical data were analyzed by Graphpad prism 8. The experiments have been conducted either three times or with a large enough number of mice to identify a statistically significant difference in the means. All values are expressed as mean ± SEM. One-way ANOVA Tukey's multiple comparisons test was used to determine statistical significance for column graph by GraphPad Prism 8.0 (GraphPad Software, Inc.). Two-way ANOVA Sidak's or Tukey's multiple comparisons test was used to determine statistical significance for grouped graph by GraphPad Prism 8.0 (GraphPad Software, Inc.). $P$ values < 0.05 were considered statistically significant.

## Graphical content

Graphical content (Figs. 8c, 9a and h) was created with BioRender.com.

## Reporting summary

Further information on research design is available in the Nature Portfolio Reporting Summary linked to this article.

## Data availability

Data supporting the findings of this work are available within the paper and its Supplementary Information files. Source data are provided with this paper.

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

## Acknowledgements

This work was supported by the National Natural Science Foundation of China (No. 82122037, K.F., No. 81930050, X.Y., No. 32301197, X.M.), National Key Research and Development Program of China (No. 2021YFC2102900, K.F.), Youth Innovation Promotion Association of Chinese Academy of Sciences (2019093, K.F.), CAS Interdisciplinary Innovation Team (JCTD-2020-08, K.F.) and Key Laboratory of Biomacromolecules, Chinese Academy of Sciences (ZGD-2023-03, K.F.).

## Author contributions

K.F, X.M. and H.F. conceived and designed the whole project. X.M. and H.F. designed and performed the experiments. L.C. and X.G. designed and performed the DFT calculation. C.H., J.X., Y.H., K.W. and L.G.

provided relevant experimental technical support and helped with the investigations. J. H. contributed to the manuscript writing. K.F., X.M., H.F., L.C. and X.Y. wrote the manuscript with input from all authors.

## Competing interests

The authors declare no competing interests.
