## [Peer Review File · Nature Communications]

Reviewers' Comments:

Reviewer #1:

Remarks to the Author:

The work reported one of the ultrasmall metal alloy nanozyme system to mimic neutrophil enzymatic cascades for tumor catalytic therapy. By changing the ratios of Au and Pd, the SOD and MPO-like activities can be therefore modulated and the optimal catalytic activity has also been explored and determined by theoretical calculations. This work has provided one simple solution to control the activities of nanozymes, as one of promising system for tumor treatment. While, there still remains some questions, what are the advantages of this MPO-like system compared to previous natural SOD-MPO cascade systems (Nature Communications, 2019, 10, 240)? Are there any specific guidelines for the design of different kind of metal-based nanozymes, if we want to construct the GOx-like activity or LOx-like activity, such like that. I recommend the acceptance of this manuscript after addressing the above and following comments.

1. From figure 2a, we can see that the Au1Pd3 showed the highest SOD activity, if there is an optimal ratio to achieve the best SOD activity?
2. Since Au1Pd3 is localized in the lysosomes, it is suggested to supplement the corresponding stability experiments in a lysosome-like environment (pH 4.5).
3. It showed that H₂O₂ levels could influence the efficacy of nanozymes, the authors should provide more information on this issue and justify it in the work.
4. The English writing and grammars in this work needs to be improved or polished by natives. For example, "High-resolution transmission electron microscopy (HRTEM) images showed that the synthesized five nanozymes exhibited a spherical morphology and had a uniform distribution with a diameter of 2-3 nm", and so on.
5. In figure.2d, the "Au1Pd1" on the axis appears to be mislabeled.

Reviewer #2:

Remarks to the Author:

This is an interesting study that explores the development of nanozymes with dual activities of SOD and MPO-like activities for a novel cancer therapy strategy that emulates the killing mechanism of neutrophils. The authors showed that by adjusting the alloy ratio, the activities of the nanozymes could be regulated. Through theoretical calculations, the authors elucidated the catalytic mechanism of the nanozymes. The authors further demonstrated that this cancer therapy strategy has outstanding efficacy and biosafety. This work sheds light on a new way to design and control the activities of nanozymes. The research is methodically organized, and the conclusions drawn from the experiments are well-substantiated by experimental data. Overall, the manuscript is clearly written; I believe that the manuscript is suitable for publication in Nature Communications after addressing the following questions.

1. "Nanozymes, as a new type of artificial enzymes, provide new materials for the development of enzyme-catalyzed therapies". The reference for this sentence does not fully support this conclusion; a more recent comprehensive reference is needed. Please make sure all the claims are well-supported with literature, this problem should be carefully checked.
2. It is not clear from the results section what the difference was for the synthesis of the five nanozymes; how were the different ratios of alloy obtained?
3. After the Pd content is determined, the five nanozymes should be assigned their names: Au and Pd, Au₃Pd₁, Au₂Pd₂ and Au₁Pd₃
4. About theoretical calculation, please supplement the reason for calculating specifically on the (111) facets rather than other facets.
5. What is the purpose of evaluating the thermal stability of two HO₂• radicals on Au₁Pd₃ (111) and Pd (111) surfaces at 300 K?
6. There are some writing errors in this manuscript. For instance, hydroxyl radical should be written as •OH instead of OH•. In Figures 8a and 8h, "eablishing" should be "establishing".

Reviewer #3:

Remarks to the Author:

In this manuscript titled "Ultrasmall metal alloy nanozymes mimicking neutrophil enzymatic cascades for tumor catalytic therapy," authors have investigated the catalytic activities of ultrasmall metal alloy nanozymes and their potential applications in biomimetic tumor therapy. The nanozymes are designed to mimic the activities of natural enzymes involved in immune-mediated killing, particularly the neutrophil's mechanism involving superoxide dismutase (SOD) and myeloperoxidase (MPO). Key results from their studies are outlined below.

- Five types of ultrasmall AuPd alloy nanozymes were synthesized: Au, Au₃Pd₁, Au₂Pd₂, Au₁Pd₃, and Pd.
- The nanozymes demonstrated both SOD-like and MPO-like activities, closely related to their alloy ratios.
- These nanozymes were found to initiate a cascade catalytic reaction, similar to the neutrophil's mechanism, producing highly oxidizing species like HClO and singlet oxygen (O₂).
- Among the nanozymes, Au₁Pd₃ exhibited the highest cascade enzymatic activity, enabling the simulation of neutrophil-like killing mechanisms for in vitro and in vivo tumor treatment.
- The study also provided theoretical calculations elucidating the catalytic mechanisms underlying the nanozymes' activities and their relationship with alloy ratios.
- Finally, the authors concluded that the biomimetic tumor therapy described in their study could pave the way for further advancements in biomimetic treatments for tumors and other diseases.

The concept of utilizing catalytic reactions to simulate immune responses offers an innovative perspective on cancer treatment strategies. The manuscript demonstrates a high level of scientific thoroughness through a well-defined experimental setup, detailed methodologies, and extensive characterization techniques. The cascade catalytic activities, including SOD and MPO-like activities, were systematically investigated, providing a solid foundation for the proposed biomimetic therapy. Further, the inclusion of theoretical calculations elucidating the catalytic mechanisms adds depth to the study and enhances understanding of the nanozymes' activities. This combination of experimental and theoretical approaches strengthens the manuscript's scientific foundation.

Therefore, I am recommending this manuscript for publication in this journal. However, I have identified some areas in the manuscript that requires further improvement:

1. For instance, in page 10 through 11, authors stated the correlation between transition metal's d-band center theory and their reactivity. This theory needs to be explicitly stated and referenced.
2. The manuscript could benefit from improved organization and clarity. Some sections appear lengthy, which might hinder readers' comprehension. Simplifying complex explanations and using for instance subfigures 3c and e (in page 12) will be better than just Fig. 3c, e and so on.
3. While the study provides significant positive results, it would be worthwhile to discuss any limitations or challenges encountered during the research. This would add transparency to the study and acknowledge areas where further investigation might be needed.
4. Also, while the study emphasizes the novelty of the proposed approach, it should also include a comprehensive discussion comparing the results with existing literature on nanozymes, catalytic therapies, and immunotherapy. This would place the study's contributions in a broader context.
5. The statistical analysis of the results could be more explicit. Detailed descriptions of the statistical tests used, p-values, and significance thresholds would enhance the robustness of the findings.
6. Lastly, the manuscript mentions that all research complied with relevant ethical regulations, but more details about ethical approval, animal welfare, and informed consent, if applicable, should be provided to ensure ethical standards are met.
7. The Pd or Au atoms have been randomly distributed on the AuPd alloy surfaces. For example, Pd monomer, which is a single Pd atom completely surrounded by the neighboring Au atoms, has been known to show the higher catalytic activity in various reactions (like O₂ reduction to H₂O₂). What is the key Pd or Au ensemble in enhancing SOD-like and MPO-like activities? In addition, the authors need to mention the stability of such Pd or Au ensembles.
8. In alloy catalysts, the ligand, strain, and ensembles effects play the important role in determining catalytic activity. Which alloy effect is the key in enhancing catalysis in this study?
9. Size of AuPd alloy catalyst may affect the catalysis. The authors should discuss this effect in manuscript.
10. Explain how to calculate the d-band center. The d-band center is the average of surface Au and Pd atoms?
11. The authors should discuss the reason why the Al₁Pd₃ alloy has higher activity than Pd by using Figure 3(d) and 3(f). I cannot see any significant activity difference for Al₁Pd₃ and Pd. Note

that the barrier difference is only 0.01eV

REVIEWER COMMENTS

Reviewer #1 (Remarks to the Author):

The work reported one of the ultras-small metal alloy nanozyme system to mimic neutrophil enzymatic cascades for tumor catalytic therapy. By changing the ratios of Au and Pd, the SOD and MPO-like activities can be therefore modulated and the optimal catalytic activity has also been explored and determined by theoretical calculations. This work has provided one simple solution to control the activities of nanozymes, as one of promising system for tumor treatment.

Response: We are grateful for the comments.

While, there still remains some questions, what are the advantages of this MPO-like system compared to previous natural SOD-MPO cascade systems (Nature Communications, 2019, 10, 240)?

Response: We appreciate the reviewer's insightful question. We have added the following information in red in the "Discussion" section on page 32 in the revised manuscript.

"While the previous study paved the way for developing strategies of mimicking neutrophil for tumor treatment, the key advantage of our MPO-like system lies in its reliance on nanoparticle-based inherent catalytic activity without the need for natural enzymes. This difference allows for several notable advantages.

Firstly, due to the dual enzymatic activities of the AuPd alloy nanozyme, our system only requires one type of nanozyme to achieve the SOD-MPO cascade reaction, making the entire system relatively simple. In contrast, the natural enzyme-based system required two types of natural enzymes and enzyme carrier.

Secondly, the use of nanomaterials in our system provides improved stability compared to natural enzymes. Nanoparticles are known for their robustness and resistance to degradation, rendering our system more durable and suitable for long-term applications.

Furthermore, the synthesis of nanoparticles can be achieved with relative ease and on a larger scale compared to the production of natural enzymes, which often requires complex and time-consuming purification processes.

Lastly, a nanozyme-based system can achieve precise control and optimization of catalytic performance through structural regulation. However, the manipulability of natural enzyme-based system in terms of activity regulation is relatively low.

These advantages underscore the potential of our MPO-like system as a promising alternative and advance in the field of catalysis and biomedical applications. "

Are there any specific guidelines for the design of different kind of metal-based nanozymes, if we want to construct the GOx-like activity or LOx-like activity, such like that.

Response: We appreciate the insightful question. While we cannot provide specific guidelines due to the varying trend of enzymatic activity in metal-based nanozymes

when constructing nanozymes with different metals, such as those with GOx-like activity or LOx-like activity; we can offer some general guidelines based on our research on the catalytic mechanism of AuPd alloy nanozymes. A high d-band center for metal surface and high adsorption energy of metal surface for enzymatic substrates contribute to a higher catalytic activity. Based on the above two factors, alloy nanozymes generally exhibit higher activity compared to monometallic nanozymes. However, the specific alloy ratio for optimal enzymatic activity depends on the specific metal and enzymatic activity.

We have added related information in red in the “Discussion” section on page 33 in the revised manuscript.

I recommend the acceptance of this manuscript after addressing the above and following comments.

Response: We thank the reviewer for endorsing the publication of our manuscript.

1. From figure 2a, we can see that the Au₁Pd₃ showed the highest SOD activity, if there is an optimal ratio to achieve the best SOD activity?

Response: We appreciate the insightful question. In order to address this issue, we further synthesized the Au_{1.3}Pd_{8.7} alloy nanozymes, with a Pd content that lies between that of the Au₁Pd₃ and Pd nanozymes, and investigated their SOD-like activity. We found that the alloy nanozymes, with an Au:Pd ratio of 1:3, demonstrate the optimal SOD-like activity.

Figure 2a showed that the SOD-like activity of either Au or Pd nanozymes is low, and alloying can increase their activity. Therefore, it is supposed that as the Pd content in the alloy increases, the SOD-like activity of AuPd alloy nanozymes should exhibit a pattern of initially increasing and then decreasing, with the corresponding peak ratio being the optimal ratio. Here, we further prepared the Au_{1.3}Pd_{8.7} alloy nanozymes, with the Pd content in the alloy determined to be 87% by the ICP-OES analysis. As shown in Supplementary Fig. S2, the Au_{1.3}Pd_{8.7} nanozymes consist of uniformly distributed small particles with a hydrated particle size of 3.8 nm, which is consistent with the other five nanozymes. The results of the SOD activity assay demonstrated that the Au_{1.3}Pd_{8.7} nanozymes possessed lower activity than the Au₁Pd₃ nanozymes, but higher activity than the Pd nanozymes. This suggests that the alloy nanozymes with an Au:Pd ratio of 1:3 exhibit the optimal SOD-like activity. We do not further consider the nanozymes with an Au:Pd ratio around 1:3 due to the limitations in the accuracy of the material synthesis method, as indicated by the error bar of the Au:Pd ratio in Fig. 1b. Additionally, the pursuit of the optimal ratio is an endless endeavor.

We have added the above results and information in red on page 9 in the revised manuscript.

Supplementary Fig. 2 The TEM image (a), hydrodynamic diameter distribution (b) and SOD-like activity (c) of the Au_{1.3}Pd_{8.7} nanozymes.

2. Since Au₁Pd₃ is localized in the lysosomes, it is suggested to supplement the corresponding stability experiments in a lysosome-like environment (pH 4.5).

Response: We appreciate the insightful question. We have chosen 0.2M HAc-NaAc buffer (pH 4.5) to mimic the acidic environment of the lysosomes and investigated the stability of the Au₁Pd₃ nanozymes over a period of 72 hours. The TEM image presented in Supplementary Fig. 9a showed that the Au₁Pd₃ nanozymes maintained their stability without experiencing aggregation during the entire experimental duration. Furthermore, Supplementary Fig. 9b demonstrated that the size of the Au₁Pd₃ nanozymes remained consistently below 6 nm throughout the observation period. These results indicate that the Au₁Pd₃ nanozymes exhibit robust stability and maintain their nanostructural integrity in a lysosome-like environment, which is the basis for the Au₁Pd₃ nanozymes to exert their enzymatic activities in lysosomes.

We have added the results and information about this issue in red on page 17 in the revised manuscript.

Supplementary Fig. 9 TEM image, picture and size distribution (a) and hydrodynamic diameter distribution (b) of Au₁Pd₃ nanozymes in a pH 4.5 lysosome-mimicking environment for 72 h. Scale bar = 10 nm.

3. It showed that H₂O₂ levels could influence the efficacy of nanozymes, the authors should provide more information on this issue and justify it in the work.

Response: We sincerely appreciate the insightful suggestion raised by the reviewer.

Nanozymes have demonstrated great potential in regulating reactive oxygen species for tumor catalytic therapy and antimicrobial applications. However, it is worth noting that the therapeutic efficacy of these nanozymes can be limited due to their low affinity for H_2O_2 and the typically low levels of H_2O_2 in the tumor microenvironment and inside microbes.

In order to address these challenges, our work focuses on leveraging a cascade catalytic reaction involving SOD and MPO. The SOD catalytic reaction can generate H_2O_2 , which serves as the substrate for the MPO catalysis. By utilizing this cascade reaction, our system overcomes the limitation of relying solely on H_2O_2 levels in the tumor microenvironment, which are typically below 0.1 mM. This eliminates the need for additional H_2O_2 administration, which not only simplifies the treatment process but also improves the feasibility of subsequent clinical translation.

We have emphasized this point in red on page 4 in our revised manuscript to justify our work.

4. The English writing and grammars in this work needs to be improved or polished by natives. For example, “High-resolution transmission electron microscopy (HRTEM) images showed that the synthesized five nanozymes exhibited a spherical morphology and had a uniform distribution with a diameter of 2-3 nm”, and so on.

Response: We appreciate the reviewer's feedback regarding the language and grammar in our work. We have addressed the linguistic issues and made necessary improvements to ensure clarity and coherence in our revised manuscript.

Regarding the specific example provided, we have revised the sentence to "High-resolution transmission electron microscopy (HRTEM) images demonstrated the spherical morphology and uniform distribution of the synthesized five nanozymes, all exhibiting a diameter range of 2-3 nm." in red on page 6 in the revised manuscript.

5. In figure.2d, the “Au1Pd1” on the axis appears to be mislabeled.

Response: We thank the reviewer for pointing out this mistake. We have corrected this label in red on page 11 in the revise manuscript as follows.

Fig. 2 (d) Production of HClO by the SOD-MPO-like cascade activity ($n = 3$ independent experiments).

Reviewer #2 (Remarks to the Author):

This is an interesting study that explores the development of nanozymes with dual activities of SOD and MPO-like activities for a novel cancer therapy strategy that emulates the killing mechanism of neutrophils. The authors showed that by adjusting the alloy ratio, the activities of the nanozymes could be regulated. Through theoretical calculations, the authors elucidated the catalytic mechanism of the nanozymes. The authors further demonstrated that this cancer therapy strategy has outstanding efficacy and biosafety. This work sheds light on a new way to design and control the activities of nanozymes. The research is methodically organized, and the conclusions drawn from the experiments are well-substantiated by experimental data. Overall, the manuscript is clearly written; I believe that the manuscript is suitable for publication in Nature Communications after addressing the following questions.

Response: We thank the reviewer for the positive comments.

1. “Nanozymes, as a new type of artificial enzymes, provide new materials for the development of enzyme-catalyzed therapies”. The reference for this sentence does not fully support this conclusion; a more recent comprehensive reference is needed. Please make sure all the claims are well-supported with literature, this problem should be carefully checked.

Response: We thank the reviewer's valuable suggestion and acknowledge the importance of providing accurate and well-supported claims in our research article. To address this concern, we have carefully reviewed and updated our references mentioned below to ensure that the claims made in the manuscript are supported by recent, relevant, and comprehensive literature.

- However, due to the fact that the chemical nature of natural enzymes is protein, they typically present poor stability and are prone to structural changes and inactivation in acidic, alkaline, and thermal environments. In addition, the extraction process of natural enzymes is complex and costly. Furthermore, there is an issue of immunogenicity in the application of natural enzymes in the body. These factors limit the clinical development and application of natural enzymes as drugs^{1,2}.
- Nanozymes, as a new type of artificial enzymes, provide new materials for the development of enzyme-catalyzed therapies^{3,4}.
- In recent years, noble metal nanozymes have attracted considerable attention due to their remarkable catalytic activity⁵.

1 Meng, X. Q., Fan, K. L. & Yan, X. Y. Nanozymes: An emerging field bridging nanotechnology and enzymology. *Science China-Life Sciences* **62**, 1543-1546 (2019).

2 De La Fuente, M. et al. Enzyme therapy: Current challenges and future perspectives. *International Journal of Molecular Sciences* **22**, (2021).

3 Wei, H. & Wang, E. K. Nanomaterials with enzyme-like characteristics (nanozymes): Next-generation artificial enzymes. *Chemical Society Reviews* **42**, 6060-6093 (2013).

4 Wu, J. J. X. et al. Nanomaterials with enzyme-like characteristics (nanozymes):

Next-generation artificial enzymes (ii). *Chemical Society Reviews* **48**, 1004-1076 (2019).

5 Liu, Q. W., Zhang, A., Wang, R. H., Zhang, Q. & Cui, D. X. A review on metal- and metal oxide-based nanozymes: Properties, mechanisms, and applications. *Nano-Micro Letters* **13**, (2021).

2. It is not clear from the results section what the difference was for the synthesis of the five nanozymes; how were the different ratios of alloy obtained?

Response: We appreciate the reviewer for pointing out this valuable issue. We have incorporated the following clarification into the revised manuscript in red on page 6: “The alloy ratio was regulated by adjusting the proportions of metal precursors.”

3. After the Pd content is determined, the five nanozymes should be assigned their names: Au and Pd, Au₃Pd₁, Au₂Pd₂ and Au₁Pd₃

Response: We appreciate the insightful suggestion and have added the following description following the ICP-OES results in the revised manuscript in red on page 6: “These nanozymes were labeled as Au, Au₃Pd₁, Au₂Pd₂, Au₁Pd₃, and Pd, corresponding to their respective alloy compositions.”

4. About theoretical calculation, please supplement the reason for calculating specifically on the (111) facets rather than other facets.

Response: We are thankful to the reviewer for this expert suggestion. We have added relative information in our revised manuscript in red on page 12 as follows: “The (111) facets of Au, Pd and their alloys are selected accounting for catalytic activities in our calculations due to their more energetically stable configurations than the other facets^{1,2}. Moreover, the (111) facets are also the predominant facets in the synthesized nanozymes, as shown by XRD patterns in Fig. 1e.”

Fig. 1e XRD patterns of five AuPd alloy nanozymes

Reference:

- 1 Tyson, W. R. & Miller, W. A. Surface free-energies of solid metals - estimation from liquid surface-tension measurements. *Surface Science* 62, 267-276 (1977).
- 2 Li, J. N., Liu, W. Q., Wu, X. C. & Gao, X. F. Mechanism of pH-switchable peroxidase and catalase-like activities of gold, silver, platinum and palladium. *Biomaterials* 48, 37-44 (2015).

5. What is the purpose of evaluating the thermal stability of two HO₂• radicals on Au₁Pd₃ (111) and Pd (111) surfaces at 300 K?

Response: We are thankful to the reviewer for this insightful question. As the product of SOD activity, the H₂O₂ on metal surface can generate two •OH radicals or a H₂O molecule and a reactive O* atom. In order to determine the final product of two HO₂• radicals on Au₁Pd₃ (111) and Pd (111) surfaces, the thermal stability at 300 K is performed within 20 ps, and the results indicate that the reactive O* atom acts as the product of two HO₂ free radicals on the metal surface. This confirms the rationality of the process from state 2 to state 3, as illustrated in Figure 3d and 3f.

We have added the above information in red on page 13 in our revised manuscript.

6. There are some writing errors in this manuscript. For instance, hydroxyl radical should be written as •OH instead of OH•. In Figures 8a and 8h, “eatablishing” should be “establishing”.

Response: We thank the reviewer for pointing out these mistakes. We have corrected them in the revise manuscript in red.

Reviewer #3 (Remarks to the Author):

In this manuscript titled “Ultrasmall metal alloy nanozymes mimicking neutrophil enzymatic cascades for tumor catalytic therapy,” authors have investigated the catalytic activities of ultrasmall metal alloy nanozymes and their potential applications in biomimetic tumor therapy. The nanozymes are designed to mimic the activities of natural enzymes involved in immune-mediated killing, particularly the neutrophil's mechanism involving superoxide dismutase (SOD) and myeloperoxidase (MPO). Key results from their studies are outlined below.

- Five types of ultrasmall AuPd alloy nanozymes were synthesized: Au, Au₃Pd₁, Au₂Pd₂, Au₁Pd₃, and Pd.
- The nanozymes demonstrated both SOD-like and MPO-like activities, closely related to their alloy ratios.
- These nanozymes were found to initiate a cascade catalytic reaction, similar to the neutrophil's mechanism, producing highly oxidizing species like HClO and singlet oxygen (O₂).
- Among the nanozymes, Au₁Pd₃ exhibited the highest cascade enzymatic activity, enabling the simulation of neutrophil-like killing mechanisms for in vitro and in vivo tumor treatment.
- The study also provided theoretical calculations elucidating the catalytic mechanisms underlying the nanozymes' activities and their relationship with alloy ratios.
- Finally, the authors concluded that the biomimetic tumor therapy described in their study could pave the way for further advancements in biomimetic treatments for tumors and other diseases.

The concept of utilizing catalytic reactions to simulate immune responses offers an innovative perspective on cancer treatment strategies. The manuscript demonstrates a high level of scientific thoroughness through a well-defined experimental setup, detailed methodologies, and extensive characterization techniques. The cascade catalytic activities, including SOD and MPO-like activities, were systematically investigated, providing a solid foundation for the proposed biomimetic therapy. Further, the inclusion of theoretical calculations elucidating the catalytic mechanisms adds depth to the study and enhances understanding of the nanozymes' activities. This combination of experimental and theoretical approaches strengthens the manuscript's scientific foundation. Therefore, I am recommending this manuscript for publication in this journal.

Response: We thank the reviewer for the positive comments.

However, I have identified some areas in the manuscript that requires further improvement:

1. For instance, in page 10 through 11, authors stated the correlation between transition metal's d-band center theory and their reactivity. This theory needs to be explicitly stated and referenced.

Response: We are thankful to the reviewer for this insightful suggestion. We have added following information in our revised manuscript in red on page 11.

“The d-band center theory states that, the d-band center for various metals is an indicator to explain the adsorption energy trends for a given adsorbate: the higher the d-states are in energy relative to the Fermi level, the more empty the anti-bonding states and the larger the adsorption energy, which has been widely used to explain the relative reactivity of metal surfaces^{1,2,3}.”

Reference:

1 Norskov, J. K., Abild-Pedersen, F., Studt, F. & Bligaard, T. Density functional theory in surface chemistry and catalysis. *Proceedings of the National Academy of Sciences of the United States of America* 108, 937-943 (2011).

2 Hammer, B. & Norskov, J. K. Why gold is the noblest of all the metals. *Nature* 376, 238-240 (1995).

3 Takigawa, I., Shimizu, K. I., Tsuda, K. & Takakusagi, S. Machine-learning prediction of the d-band center for metals and bimetals. *Rsc Advances* 6, 52587-52595 (2016).

2. The manuscript could benefit from improved organization and clarity. Some sections appear lengthy, which might hinder readers' comprehension. Simplifying complex explanations and using for instance subfigures 3c and e (in page 12) will be better than just Fig. 3c, e and so on.

Response: We appreciate the reviewer's valuable feedback on the organization and clarity of our manuscript. To address the issue, we have carefully reorganized the manuscript to improve its flow and structure. Moreover, we took the reviewer's suggestion to simplify complex explanations into consideration.

Regarding the references to specific figures, we have revised the manuscript according to the reviewer's suggestion and the journal's formatting instructions in red on page 13.

3. While the study provides significant positive results, it would be worthwhile to discuss any limitations or challenges encountered during the research. This would add transparency to the study and acknowledge areas where further investigation might be needed.

Response: We are grateful to the reviewer for the valuable suggestion. Indeed, during our research process, we encountered a few limitations.

The first challenge is the issue of by-products in the synthetic process of ultrasmall AuPd alloy nanozymes. It is crucial to ensure that the studied AuPd alloy nanozymes are of ultrasmall size, as the ultrasmall dimension of alloy nanozymes contributes to their high enzymatic activities (which will be discussed in detail in question 9) and their

ability to be cleared by the kidneys. However, not all products obtained in the synthesis process possess the desired ultrasmall size. In reality, some larger particles are formed as byproducts, thus necessitating the use of ultrafiltration tubes to eliminate these particles and guarantee the uniformity of the ultrasmall nanozymes. Consequently, it would be worthwhile to further optimize the method employed for ultrasmall nanozyme synthesis, with the aim of simplifying the process and enhancing the overall yield.

The second challenge in the research process is how to achieve the renal clearance ability and tumor targeting ability of the ultrasmall Au₁Pd₃ nanozymes simultaneously. The ultrasmall size of the nanozymes enables them rapid metabolism by the kidneys. However, this also leads to limited accumulation of the nanozymes at the tumor site. To address this issue, we introduced a tumor-targeting molecule, folic acid (FA), to modify the ultrasmall Au₁Pd₃ nanozymes. Despite this modification, half of the nanozymes were still eliminated from the body through urine, as depicted in Fig. 6d. Therefore, further investigation is necessary to strike a balance between the renal clearance and tumor targeting abilities. For instance, optimizing the modification density of tumor-targeting molecules or exploring alternative tumor-targeting molecules are both worth considering.

We have added the above information in the “Discussion” section in red on page 33 in the revised manuscript.

Fig. 6d Pd amounts in urine and feces at various intervals from CT26 tumor-bearing mice after intravenous injection of Au₁Pd₃-FA-Cy5.5 nanozymes (n = 3 mice).

4. Also, while the study emphasizes the novelty of the proposed approach, it should also include a comprehensive discussion comparing the results with existing literature on nanozymes, catalytic therapies, and immunotherapy. This would place the study's contributions in a broader context.

Response: We appreciate the reviewer's constructive suggestion and would like to address the raised points.

When it comes to immunotherapy¹, it entails artificially enhancing or inhibiting the body's immune function to treat diseases. Immunotherapy primarily focuses on bolstering the body's own immune system rather than directly targeting the diseased tissue. Although our work involves simulating the killing mechanism of neutrophils

using nanozymes, our focus is directly targeting the tumor sites with this killing effect, rather than regulating the body's immune system. Therefore, it does not fall within the scope of immunotherapy and we will refrain from comparing our study with immunotherapy in the manuscript. Instead, we primarily compare our results with previous works on nanozyme-based therapies and catalytic therapies.

Nanozymes have demonstrated great potential in regulating ROS for tumor catalytic therapy. However, it is worth noting that the therapeutic efficacy of these nanozymes can be limited due to their low affinity for H₂O₂ and the typically low levels of H₂O₂ in the tumor microenvironment². At present, most effective nanozyme-based tumor therapy strategies are carried out in combination with additional therapeutic interventions, including photothermal therapy³, photodynamic therapy⁴, sonodynamic therapy⁵, and immunotherapy⁶. In contrast, our work addresses these challenges by leveraging the cascade catalytic reaction of SOD and MPO. The SOD catalytic reaction can generate H₂O₂, which serves as the substrate for the MPO catalysis. By utilizing this cascade reaction, our system overcomes the limitation of relying solely on H₂O₂ levels in the tumor microenvironment, which are typically below 0.1 mM. This is mainly due to the cascade reaction resulting from the dual-enzymatic activities of nanozymes, which not only simplifies the treatment process but also improves the feasibility of subsequent clinical translation.

Catalytic therapy associated with this article typically refers to natural enzyme-mediated catalytic therapy. While a previous study utilized natural enzymes to develop a tumor therapy strategy by mimicking the killing mechanism of neutrophils⁷, the key advantage of our nanozyme-based system lies in its reliance on nanoparticle-based inherent catalytic activity without the need for natural enzymes. This difference allows for several notable advantages^{8,9}. Firstly, due to the dual enzymatic activities of the AuPd alloy nanozyme, our system only requires one type of nanozyme to achieve the SOD-MPO cascade reaction, making the entire system relatively simple. In contrast, the natural enzyme-based system required two types of natural enzymes and enzyme carrier. Secondly, the use of nanomaterials in our system provides improved stability compared to natural enzymes. Nanoparticles are known for their robustness and resistance to degradation, rendering our system more durable and suitable for long-term applications. Furthermore, the synthesis of nanoparticles can be achieved with relative ease and on a large scale compared to the production of natural enzymes, which often requires complex and time-consuming purification processes. Lastly, a nanozyme-based system can achieve precise control and optimization of catalytic performance through structural regulation. However, the manipulability of natural enzyme-based system in terms of activity regulation is relatively low. These advantages underscore the potential of our nanozyme-based system as a promising alternative and advance in the field of catalysis and biomedical applications.

We have included the above information in the “Introduction” and “Discussion” sections in red on pages 4 and 32 in the revised manuscript, which has helped to strengthen the manuscript's overall context and significance.

Reference:

- 1 Couzin-Frankel, J. Cancer immunotherapy. *Science* **342**, 1432-1433 (2013).
- 2 Meng, X. Q. et al. High-performance self-cascade pyrite nanozymes for apoptosis-ferroptosis synergistic tumor therapy. *ACS Nano* **15**, 5735-5751 (2021).
- 3 Li, S. S. et al. A nanozyme with photo-enhanced dual enzyme-like activities for deep pancreatic cancer therapy. *Angewandte Chemie-International Edition* **58**, 12624-12631 (2019).
- 4 Zeng, Z. L. et al. Activatable polymer nanoenzymes for photodynamic immunometabolic cancer therapy. *Advanced Materials* **33**, 2021).
- 5 Zhong, X. Y. et al. Gsh-depleted ptcu3 nanocages for chemodynamic- enhanced sonodynamic cancer therapy. *Advanced Functional Materials* **30**, 2020).
- 6 Wen, M. et al. Artificial enzyme catalyzed cascade reactions: Antitumor immunotherapy reinforced by nir-ii light. *Angewandte Chemie-International Edition* **58**, 17425-17432 (2019).
- 7 Wu, Q. et al. Cascade enzymes within self-assembled hybrid nanogel mimicked neutrophil lysosomes for singlet oxygen elevated cancer therapy. *Nature Communications* **10**, (2019).
- 8 De La Fuente, M. et al. Enzyme therapy: Current challenges and future perspectives. *International Journal of Molecular Sciences* **22**, (2021).
- 9 Wei, H. & Wang, E. K. Nanomaterials with enzyme-like characteristics (nanozymes): Next-generation artificial enzymes. *Chemical Society Reviews* **42**, 6060-6093 (2013).

5. The statistical analysis of the results could be more explicit. Detailed descriptions of the statistical tests used, p-values, and significance thresholds would enhance the robustness of the findings.

Response: We thank the reviewer for the valuable suggestion. We have included relative descriptions of statistical analysis in red in the revised manuscript, as recommended by the reviewer and in accordance with the formatting guidelines of the journal. The specific statistical tests used for each result were explained in the figure legend. The exact p-values were indicated on the graph. The overall statistical analysis methods including significance thresholds were described in the "Methods" section on page 43 as follows:

Statistical analysis

General statistical data were analyzed by Graphpad prism 8. Experiments have been repeated three times or had a sufficient number of mice to detect a statistically significant difference in the means. All values are expressed as mean \pm SEM. One-way ANOVA Tukey's multiple comparisons test was used to determine statistical significance for column graph by GraphPad Prism 8.0 (GraphPad Software, Inc.). Two-way ANOVA Sidak's or Tukey's multiple comparisons test was used to determine statistical significance for grouped graph by GraphPad Prism 8.0 (GraphPad Software, Inc.). *P* values <0.05 were considered statistically significant.

6. Lastly, the manuscript mentions that all research complied with relevant ethical regulations, but more details about ethical approval, animal welfare, and informed

consent, if applicable, should be provided to ensure ethical standards are met.

Response: We are grateful to the reviewer for the valuable suggestion. We have included details about this issue in red on page 34 in the revised manuscript as follows: The animal studies were conducted in accordance with the approved protocol of the Institutional Animal Care and Use Committee (IACUC) of the Institute of Biophysics, Chinese Academy of Sciences (Project number: SYXK2023168). 6-8-week-old female BALB/c mice were purchased from Spiff (Beijing) Biotechnology Co., Ltd. All mice were group-housed 5 mice per cage in a specific pathogen-free environment in temperature (22-26°C) and humidity (40%-70%) house rooms on a 12 h light, 12 h dark cycle. The maximal tumor size permitted by the IACUC is 15 mm in diameter, in our work no mice exceeded this criterion. Following the IACUC guidelines, weight loss of more than 20%, or mice exhibiting signs of hunched posture, impaired locomotion or respiratory distress are criteria followed for prompt euthanasia by CO₂ gas. Otherwise, the mice were euthanized until the end of the experiment.

7. The Pd or Au atoms have been randomly distributed on the AuPd alloy surfaces. For example, Pd monomer, which is a single Pd atom completely surrounded by the neighboring Au atoms, has been known to show the higher catalytic activity in various reactions (like O₂ reduction to H₂O₂). What is the key Pd or Au ensemble in enhancing SOD-like and MPO-like activities? In addition, the authors need to mention the stability of such Pd or Au ensembles.

Response: We thank the reviewer for the valuable question and suggestion. In our study, both the experimental results and theoretical calculations indicate that the alloy structure of Au₁Pd₃ is the most effective to achieve SOD-MPO-like cascade catalysis. From the XRD pattern of Au₁Pd₃ nanozymes (Fig. 1e), it can be seen that face-centered cubic (111) facets are the predominant facets, corresponding to the uniform dispersion of a single Au atom surrounded by 6 Pd atoms of the top layer in Au₁Pd₃ (111) surface, as present in the Figure only for reviewer 1.

Fig. 1e XRD patterns of five AuPd alloy nanozymes.

Figure only for reviewer 1. The key ensembles in enhancing SOD-like and MPO-like activities.

Considering the impact of ensemble stability on enzymatic activities, we directly assess the changes of SOD and MPO-like activities of Au₁Pd₃ alloy nanozymes to determine the stability of the ensemble. We conducted an incubation study with the Au₁Pd₃ alloy nanozymes at various temperatures for 2 hours and measured the changes in their SOD-like and MPO-like activities. As depicted in Supplementary Fig. 5, the variations in enzymatic activity remained within 10%, indicating the excellent stability of such ensembles.

Supplementary Fig. 5 The SOD-like activity (a) and MPO-like activity of the Au₁Pd₃ alloy nanozymes after incubation at different temperatures for 2 hours.

8. In alloy catalysts, the ligand, strain, and ensembles effects play the important role in determining catalytic activity. Which alloy effect is the key in enhancing catalysis in this study?

Response: We appreciate the reviewer's insightful question. It is suggested that, ensemble effect that is related to the Au: Pd ratio of nanozymes, is the key to enhancing catalysis in this study.

In fact, the effects of ligand, strain, and ensemble on AuPd alloy catalysts have been investigated from the perspective of adsorbate adsorption. J. K. Nørskov's group studied the adsorption of adsorbates on three kinds of Au/Pd (111) alloy surfaces (Au₁Pd₃ (111),

Au₂Pd₃ (111) and Au₃Pd₁ (111)) and showed that the contribution of the ligand effect to the adsorption energy is considerably less than that of the ensemble effect which is related to the Au: Pd ratio¹. Graeme Henkelman's group also found that the ensemble effect more significantly tunes the adsorbate binding as compared to the ligand and strain effects on AuPd (111) surface².

In our study, the theoretical calculation results demonstrated that the ratio of the two metals in AuPd alloy nanozymes significantly affect the d-band center (Fig. 3a) and the adsorption energy of HO₂• free radicals on (111) facets of Au, Pd and their alloys (Table S1), thus influencing the catalytic performance of the nanozymes (Fig. 2). The Au₁Pd₃ nanozymes exhibit the highest cascade activity, attributing to the high d-band center and adsorption energy for HO₂• free radicals.

Based on the reported researches, our experimental results, and theoretical calculations, it is suggested that, ensemble effect that is related to the Au: Pd ratio of nanozymes, is the key to enhancing catalysis in this study. The ligand and strain effects may also have influence on the electronic structure, bond lengths of the AuPd alloys and the adsorption of substrates², thus affecting the catalytic performance of the nanozymes. The more in-depth study about the alloy effect in ultrasmall AuPd alloy nanozymes will be the subject of our future research.

We have added relative information and indicated that more in-depth study is worth in the further work in the “Discussion” section in red on page 34 in the revised manuscript.

Fig. 3a D-band centers of (111) facets of Au, Pd and their alloys.

Supplementary Table 1 The adsorption energy of HO₂• free radicals on (111) facets of Au, Pd and their alloys, the red value is the adsorption energy in the lowest energetic conformation.

	top	bridge	fcc	hcp
Au (111)	- 0.46	- 0.47	- 0.42	- 0.42
Au ₃ Pd ₁ (111)	- 0.96		- 0.76	- 0.66
Au ₂ Pd ₂ (111)	- 1.15	- 1.00	- 1.00	
Au ₁ Pd ₃ (111)	- 1.61	- 1.59	- 1.49	- 1.53
Pd (111)	- 1.30	- 1.22	- 1.17	- 1.18

Fig. 2 SOD-like and MPO-like activities of five AuPd alloy nanozymes. (a) SOD-like and (b) MPO-like activity of five AuPd alloy nanozymes ($n = 3$ independent experiments). (c) Method for detecting the SOD-MPO-like cascade activity. (d) Production of HClO and (e) $^1\text{O}_2$ by the SOD-MPO-like cascade activity ($n = 3$ independent experiments).

Reference:

- Liu, P. & Norskov, J. K. Ligand and ensemble effects in adsorption on alloy surfaces. *Physical Chemistry Chemical Physics* **3**, 3814-3818 (2001).
- Li, H., Shin, K. & Henkelman, G. Effects of ensembles, ligand, and strain on adsorbate binding to alloy surfaces. *Journal of Chemical Physics* **149**, (2018).

9. Size of AuPd alloy catalyst may affect the catalysis. The authors should discuss this effect in manuscript.

Response : We appreciate the insightful suggestion. To address this issue, we compared the SOD-like and MPO-like activities of AuPd alloy nanozymes with different sizes and found that size affects the enzymatic performance of nanozymes, and the smaller-sized nanozymes exhibited higher activity.

As mentioned in the question 3, in the synthesis process of ultrasmall AuPd alloy, there will be the generation of byproduct large-sized AuPd alloy nanoparticles. We use ultrafiltration tubes with cutoff 10 kDa and 100 kDa to separate them and obtain AuPd alloy nanoparticles with ultrasmall (demonstrated in the manuscript) and large sizes for studying the influence of size on catalysis. HRTEM image showed that the large nanozymes also exhibit a spherical morphology and have a uniform distribution with an average diameter of 6.4 nm (Supplementary Fig. 3a). The average hydrodynamic diameter of the large Au₁Pd₃ alloy nanoparticles is 19.2 nm (Supplementary Fig. 3b). Assays were conducted on the SOD-like and MPO-like activities of these two sizes of Au₁Pd₃ nanozymes, and it was observed that both activities of the large Au₁Pd₃ alloy were lower than that of the ultrasmall Au₁Pd₃ alloy nanozymes (Supplementary Fig. 3c

and d). This size effect is because, under the same molar or mass concentration of Au₁Pd₃ alloy, ultrasmall Au₁Pd₃ nanozymes have a larger specific surface area, which provides more active sites for the reaction. Similar size-dependent catalytic performances were reported by other researchers^{1,2}. For instance, Fan and colleagues documented that the glucose oxidase-mimetic performance of Au nanoparticles was inversely related to the size.

In conclusion, through studying the activity of Au₁Pd₃ alloys in two different sizes, we discovered that the smaller the catalyst size, the higher the catalytic activity, which may be related to the higher specific surface area and more exposed active sites. Therefore, the high SOD-like and MPO-like activities of the ultrasmall AuPd alloy nanozymes in the manuscript also benefit from their ultra-small size.

We have added relative results and information in red on page 10 in the revised manuscript.

Supplementary Fig. 4 The TEM image and size distribution (a), hydrodynamic diameter distribution (b), SOD-like activity (c) and MPO-like activity (d) of large-sized Au₁Pd₃ alloy nanozymes. Scale bar = 10 nm.

10. Explain how to calculate the d-band center. The d-band center is the average of surface Au and Pd atoms?

Response: We extend our gratitude to the reviewer for this valuable suggestion. In our calculations, the d-band center for the (111) facets of Au, Pd and their alloys can be determined using the following formula:

$$\varepsilon_d = \frac{\int_{-\infty}^{\infty} n_d(\varepsilon) \varepsilon d\varepsilon}{\int_{-\infty}^{\infty} n_d(\varepsilon) d\varepsilon}$$

where ε is the d-band energy and $n(\varepsilon)$ is the d-band density. In this article, the d-band center of (111) facets is the average of surface Au and Pd atoms. Our calculations indicate that, for Au, Pd and their alloys, the change of the d-band center of the (111) facets is attributed to the variations of surface Au and Pd atoms.

We have added the above calculation method in red on page 38 in the revised manuscript.

11. The authors should discuss the reason why the Al1Pd3 alloy has higher activity than Pd by using Figure 3(d) and 3(f). I cannot see any significant activity difference for Al1Pd3 and Pd. Note that the barrier difference is only 0.01 eV

Response: We are grateful to the reviewer for this insightful suggestion. In fact, it is the binding energy between HO₂• and Au₁Pd₃ (111) or Pd (111) surface rather than the energy barrier that causes a significant difference in enzymatic performance. From Fig. 2, it can be seen that the SOD-like activity trend for five nanozymes is consistent with the trend of SOD-MPO-like cascade activity, indicating that SOD-like activity is the key to the overall cascade activity. Compared to the Pd (111) surface, the binding energy between the HO₂• radical and the Au₁Pd₃ (111) surface is much lower (-1.61 eV vs. -1.30 eV), indicating that the Au₁Pd₃ (111) surface has a stronger ability to capture HO₂• free radicals. In the Michaelis-Menten equation, K_M is the dissociation constant of the reaction in the following eq 3:

where the ΔG_m° is the change of standard Gibbs free energy. According to the van't Hoff equation,

$$K_M^\circ = e^{\Delta G_m^\circ / RT} \quad (6)$$

where K_M° represents the standard dissociation constant, R represents the gas constant, and T represents the temperature. According to the eq 6^{1,2}, the ratio of the two K_M° values of the SOD-MPO-like cascade catalytic cycles on Pd (111) and Au₁Pd₃ (111) surfaces was about 10⁴:1, which indicated that the SOD-MPO-like cascade catalysis was much easier to occur on the Au₁Pd₃ (111) surface than on the Pd (111) surface. This computational result was in general agreement with the experimental result.

We have highlighted the above information in red on page 14 in our revised manuscript.

Fig. 2 SOD-like and MPO-like activities of five AuPd alloy nanozymes. (a) SOD-like and **(b)** MPO-like activity of five AuPd alloy nanozymes ($n = 3$ independent experiments). **(c)** Method for detecting the SOD-MPO-like cascade activity. **(d)** Production of HClO and **(e)** $^1\text{O}_2$ by the SOD-MPO-like cascade activity ($n = 3$ independent experiments).

Reference:

- 1 Meng, X. Q. et al. High-performance self-cascade pyrite nanozymes for apoptosis-ferroptosis synergistic tumor therapy. *ACS Nano* 15, 5735-5751 (2021).
- 2 Gao, W. H. et al. Deciphering the catalytic mechanism of superoxide dismutase activity of carbon dot nanozyme. *Nature Communications* 14, (2023).

Reviewers' Comments:

Reviewer #1:

Remarks to the Author:

we are satisfied with the revision done by Fan.

Reviewer #2:

Remarks to the Author:

The authors have made very essential correction on the manuscript, the contents are clear and solid, I have no further comments. The overall quality has also been well-improved, therefore, I think this version can be accepted now.

Reviewer #3:

Remarks to the Author:

The authors resolved the issues raised by the reviewer. In AuPd alloy, the surface ensembles effect is key to the determination of catalytic activity. Thus, the authors need to add the explanation of ensemble effect in the result and discussion in main manuscript with the following reference papers of ensemble effects.

1. Liu, P. & Norskov, J. K. Ligand and ensemble effects in adsorption on alloy surfaces. *Physical Chemistry Chemical Physics* 3, 3814-3818 (2001).
2. Li, H., Shin, K. & Henkelman, G. Effects of ensembles, ligand, and strain on adsorbate binding to alloy surfaces. *Journal of Chemical Physics* 149, (2018).
3. Ham H.C., Hwang Gyeong S., Han J., Nam S. W., Lim T.H. On the Role of Pd Ensembles in Selective H₂O₂ Formation on PdAu Alloys, *The Journal of Physical Chemistry C*, 113(30), 12943-12945
4. Jakub S. Jirkovský, Itai Panas, Elisabet Ahlberg, Matej Halasa, Simon Romani, and David J. Schiffrin, Single Atom Hot-Spots at Au-Pd Nanoalloys for Electrocatalytic H₂O₂ Production, *J. Am. Chem. Soc.*, 133(48). 19432-19441

REVIEWERS' COMMENTS

Reviewer #1 (Remarks to the Author):

we are satisfied with the revision done by Fan.

Response: We are grateful for the positive comments.

Reviewer #2 (Remarks to the Author):

The authors have made very essential correction on the manuscript, the contents are clear and solid, I have no further comments. The overall quality has also been well-improved, therefore, I think this version can be accepted now.

Response: We thank the reviewer for endorsing the acceptance of our manuscript.

Reviewer #3 (Remarks to the Author):

The authors resolved the issues raised by the reviewer. In AuPd alloy, the surface ensembles effect is key to the determination of catalytic activity. Thus, the authors need to add the explanation of ensemble effect in the result and discussion in main manuscript with the following reference papers of ensemble effects.

1. Liu, P. & Norskov, J. K. Ligand and ensemble effects in adsorption on alloy surfaces. *Physical Chemistry Chemical Physics* 3, 3814-3818 (2001).
2. Li, H., Shin, K. & Henkelman, G. Effects of ensembles, ligand, and strain on adsorbate binding to alloy surfaces. *Journal of Chemical Physics* 149, (2018).
3. Ham H.C., Hwang Gyeong S., Han J., Nam S. W., Lim T.H. On the Role of Pd Ensembles in Selective H₂O₂ Formation on PdAu Alloys, *The Journal of Physical Chemistry C*, 113(30), 12943-12945
4. Jakub S. Jirkovský, Itai Panas, Elisabet Ahlberg, Matej Halasa, Simon Romani, and David J. Schiffrin, Single Atom Hot-Spots at Au–Pd Nanoalloys for Electrocatalytic H₂O₂ Production, *J. Am. Chem. Soc.*, 133(48). 19432-19441

Response: We appreciate the reviewer's insightful suggestion. We have added relative information in the “Results” section in red on page 9 and 12 and in the “Discussion” section in red on page 23 in the revised manuscript as follows.

In the “Results” section (page 9):

In AuPd alloy catalysts, the ligand, strain, and ensemble effects play vital roles in determining catalytic activity. The ensemble effect refers to modifications in the surface properties that occur due to a direct alteration in the atomic ensemble constituents. The ligand and strain effects describe the tuning of the surface electronic structure in a specific surface ensemble and the changes in bond lengths of catalysts due to differences in the lattice constants of the components, respectively. Previous research

has indicated that the ensemble effect plays a significant role in the selectivity of catalytic reactions^{1,2}. Furthermore, the ensemble effect exerts a greater influence on the adsorbate binding on the AuPd surface, surpassing the effects of ligand and strain^{3,4}. Our experimental results demonstrated that the ensemble effect that is related to the ratio of the two metals in AuPd alloy nanozymes significantly affected their SOD-MPO-like cascade catalysis. Subsequently, we performed density functional theory (DFT) calculations to investigate the catalytic mechanism.

Reference:

1. Ham H.C., Hwang Gyeong S., Han J., Nam S. W., Lim T.H. On the Role of Pd Ensembles in Selective H₂O₂ Formation on PdAu Alloys, *The Journal of Physical Chemistry C*, 113(30), 12943-12945
2. Jakub S. Jirkovský, Itai Panas, Elisabet Ahlberg, Matej Halasa, Simon Romani, and David J. Schiffrin, Single Atom Hot-Spots at Au–Pd Nanoalloys for Electrocatalytic H₂O₂ Production, *J. Am. Chem. Soc.*, 133(48). 19432-19441
3. Liu, P. & Norskov, J. K. Ligand and ensemble effects in adsorption on alloy surfaces. *Physical Chemistry Chemical Physics* 3, 3814-3818 (2001).
4. Li, H., Shin, K. & Henkelman, G. Effects of ensembles, ligand, and strain on adsorbate binding to alloy surfaces. *Journal of Chemical Physics* 149, (2018).

In the “Results” section (page 12):

The above results demonstrated that the ratio of the two metals in AuPd alloy nanozymes significantly affect the d-band center and the adsorption energy of HO₂• free radicals on (111) facets of Au, Pd and their alloys, thus influencing the catalytic performance of the nanozymes. The Au₁Pd₃ nanozymes exhibit the highest cascade activity, attributing to the high d-band center and adsorption energy for HO₂• free radicals. Drawing on the experimental findings and theoretical calculations, it is suggested that the ensemble effect, specifically linked to the Au: Pd ratio of nanozymes, is the key in enhancing catalysis in this study.

In the “Discussion” section (page 23):

Regarding the alloy effect in determining the catalytic activity of ultrasmall AuPd alloy nanozymes, our experimental and theoretical calculation results demonstrated that the ensemble effect that is related to the ratio of the two metals in AuPd alloy nanozymes significantly affect the catalytic performance of the nanozymes. However, the ligand and strain effects may also have influence on the electronic structure, bond lengths of the AuPd alloys, thus affecting their catalytic activity. The more in-depth study about the alloy effect in ultrasmall AuPd alloy nanozymes will be the subject of our future research.